# A Complete Decomposition of KL Error using Refined Information and Mode Interaction Selection

**James Enouen**[1]
*University of Southern California*
*Los Angeles, CA 90089-2905, USA*

*enouen@usc.edu*

**Mahito Sugiyama**
*National Institute of Informatics, SOKENDAI*
*Chiyoda City, Tokyo 101-8430, Japan*

*mahito@nii.ac.jp*

**Reviewed on OpenReview:** *https://openreview.net/forum?id=DxUXI15C38*

## Abstract

The *log-linear model* has received a significant amount of theoretical attention in previous decades and remains the fundamental tool used for learning probability distributions over discrete variables. Despite its large popularity in statistical mechanics and high-dimensional statistics, the majority of related energy-based models only focus on the two-variable relationships, such as Boltzmann machines and Markov graphical models. Although these approaches have easier-to-solve structure learning problems and easier-to-optimize parametric distributions, they often ignore the rich structure which exists in the *higher-order interactions* between different variables. Using more recent tools from the field of information geometry, we revisit the classical formulation of the log-linear model with a focus on higher-order mode interactions, going beyond the 1-body modes of independent distributions and the 2-body modes of Boltzmann distributions. This perspective allows us to define a *complete decomposition of the KL error*. This then motivates the formulation of a sparse selection problem over the set of possible mode interactions. In the same way as sparse graph selection allows for better generalization, we find that our learned distributions are able to more efficiently use the finite amount of data which is available in practice. We develop an algorithm called MAHGenTa which leverages a novel Monte-Carlo sampling technique for energy-based models alongside a greedy heuristic for incorporating statistical robustness. On both synthetic and real-world datasets, we demonstrate our algorithm's effectiveness in maximizing the log-likelihood for the generative task and also the ease of adaptability to the discriminative task of classification.

## 1 Introduction

Distribution learning is a fundamental task in machine learning and statistics, with applications ranging from generative tasks like density estimation and generative modeling all the way to discriminative regression tasks and unsupervised clustering tasks. This fundamental problem remains at the heart of many supervised decision problems and as a cornerstone of unsupervised learning and knowledge discovery. Probability distributions parametrized by exponential families are a representative class widely chosen for distribution modeling. Although already covering a wide variety of classical distributions for continuous variables (Gaussian, exponential, gamma, etc.), for finite and discrete variables, the hierarchical log-linear model completely describes all positive distributions over the space. Also called an energy-based model, it has remained the de facto choice of model for learning over a discrete feature space for decades and has amassed considerable

---

[1]Work done while at National Institute of Informatics.

attention over the years (Ackley et al., 1985; Sejnowski, 1986; Lee et al., 2006; Wainwright et al., 2006; Shpitser et al., 2013; Van Haaren & Davis, 2012; Lowd & Davis, 2010; Nyman et al., 2014; Højsgaard, 2004).

Despite this significant amount of work to date, the vast majority of existing approaches only deal with bivariate correlations or two-body interactions, prototypical examples being Boltzmann machines and Markov graphical models (Ackley et al., 1985; Dempster, 1972; Buhl, 1993). Although this assumption is natural to forc onto continuous variables by making the simplifying Gaussian assumption, this restriction is too severe for most real-world data distributions and is not necessary for dealing with the case of discrete variables. Many existing amendments to these approaches like maximal cliques, chordal graphs, and stratified graphical models (Shpitser et al., 2013; Nyman et al., 2014; Højsgaard, 2004) are still only graph-based or two-body structure approximations, remaining limited in their ability to describe the underlying higher-order structures which can exist within data. In this work, we instead offer a more unified perspective which further includes the hypergraphical structure encoded by higher-order interactions.

By replacing the (two-dimensional) edge graph between features with the higher-order hypergraph, we introduce a structure learning problem which is seemingly even more challenging than the usual graphical approaches. Despite this greater complexity a priori, using recent developments in the field of information geometry (Ghalamkari et al., 2023; Sugiyama et al., 2018) allows us to construct a complete decomposition of a distribution's information content, instead actually providing a greater fine-grained understanding into the structure of a probability distribution. By associating the hypergraph of the hierarchical log-linear model of a discrete distribution with the partially-ordered set (poset) of mode interactions in the corresponding probability tensor, we are able to devise a higher-order definition of non-negative information which provides a complete decomposition of the KL error for a given probability distribution.

Altogether, we demonstrate that this alternate perspective is theoretically well-supported, allowing us to define more fine-grained measurements of the information between variables and opening up new opportunities for the study of higher-order structure. Additionally, we provide practically useful learning algorithms for both the combinatorial structure and the parametric value of the distribution. We summarize our contributions as the following:

- We first define the **'refined information'** of a set of two or more variables, generalizing the mutual information of a set of two variables in a way which always returns a positive quantity measuring the information content. We show that this yields a complete decomposition of the KL error with applications in structure discovery.

- We provide the first theoretical underpinnings for the better generalization properties of higher-order Boltzmann machines via the problem of **'mode interaction selection'**, showing how to yield better sample complexity for real-world scenarios with finite datasets. We then show how the combinatorially large space of all possible interaction hierarchies can be effectively tackled by a greedy approach.

- We finally develop our model called the **M**ode-**A**ttributing **H**ierarchy for **Gen**erating **Ta**bular data (**MAHGenTa**) which implements a GPU-based gradient descent algorithm to efficiently learn the hierarchical log-linear model on both synthetic and real-world datasets. We further demonstrate how such energy-based models trained to achieve good generative performance will have automatically emergent capabilities in discriminative tasks like classification, paralleling the wide success of generative pretraining.

## 2 Background

Before introducing the main task of distribution learning, we first review the modern approaches from feature selection which provide the high-dimensional intuition for our statistical arguments and the information geometric approaches to tensor decomposition which inspires the main distribution-tensor correspondence.

### 2.1 Feature Selection and Feature Interaction Selection

Feature selection (FS) has long been a staple of machine learning for dealing with high-dimensional data, prescreening a large number of features to remove both irrelevant and redundant features from the input before the training a predictive model. This provides many benefits like reducing overfitting, faster training,

and better understanding of the data structure. Historical approaches determine relevant, irrelevant, and redundant features by understanding the mutual information between the inputs and the target variable (Shannon, 1948; McGill, 1954), whereas modern feature selection approaches concern themselves with giving the proper credence to higher-order interactions and correlations during selection, generally called 'feature-interaction-aware feature selection' (Zeng et al., 2015; Nakariyakul, 2018; Chen et al., 2015; Bennasar et al., 2015).

In recent years, however, there has been a parallel interest in feature interactions via a more general problem than feature selection called feature interaction selection (FIS) (Fan et al., 2016; Sugiyama & Borgwardt, 2019; Enouen & Liu, 2022; Lyu et al., 2023). This procedure further specifies how feature combinations are allowed to interact in the final model. For example, in a random forest model, feature interaction selection would dictate how each different tree can only use a specific interaction subset, rather than any possible subset out of the selected features. Although FIS raises the combinatorial complexity of the search problem from all possible subsets to all possible collections of subsets, the finer-grained structure further amplifies the same typical benefits of feature selection: reduced overfitting, faster training, and better understanding. In this work, we will apply this same methodology to the generative task of distribution learning instead of to the discriminative task of classification.

## 2.2 Non-Negative Tensor Decomposition

Recent works in tensor decomposition have been able to avoid the optimization difficulties of typical low-rank decompositions by instead focusing on non-negative tensors and replacing the squared error loss with the KL divergence error (Aswani, 2016; Sugiyama et al., 2018; Ghalamkari et al., 2023). Previous works assuming that a full decomposition is feasible (Sugiyama et al., 2018) or that a tensor's modes can be partitioned into independent components (Aswani, 2016) have recently been replaced by specific control over the 'many-body interactions' within the tensor (Ghalamkari et al., 2023). Following this notation, using one-body interactions corresponds to independent distributions and using two-body interactions corresponds to Boltzmann machines.

Following this correspondence between the problems of approximating a non-negative probability tensor and learning a discrete distribution via minimizing the same KL-divergence objective, we adopt the language 'mode interaction' to describe variables interacting during generative modeling, instead of the more popular 'feature interaction' of Section 2.1 which focuses on the discriminative cases. In this work, we leverage the consequent connections to information geometry (Amari, 2016) to develop a procedure of mode interaction selection (MIS) which parallels the higher-order FIS selection problem, but applied to the variable interactions between tensor modes.

## 2.3 Distribution Learning

**The Log-Linear Model** The log-linear model has always been a fundamental tool of statistics, with its use dating as far back as the historical works of Fisher (Fisher, 1934). In continuous variables, focusing on exponential families of distributions may limit us to certain types distributions (Gaussians, Poissons, etc.); however, in the case of finite variables, the log-linear model has no such limitations. Accordingly, the log-linear model has been a staple of describing any categorical distribution, being the focus of many Bayesian optimal inference frameworks as well as a central tool of statistical physics.

In the previous decade, this method received a significant amount of attention alongside the wave of sparsity methods enabled by LASSO (Tibshirani, 1996). This primarily led to an abundance of graph-based approaches which perform selection over the pairs of 2-body correlators between features, such as Markov graphs with L1 regularization (Lee et al., 2006; Wainwright et al., 2006; Lowd & Davis, 2010; Van Haaren & Davis, 2012). This was later followed up with specific types of graphical assumptions which can further simplify the learning problem (Shpitser et al., 2013; Nyman et al., 2014; Højsgaard, 2004; Massam et al., 2009). These graphical models have allowed for more fine-grained control over the structure of the learned distribution compared with previous approaches, receiving the benefits of sparsity which allows for easier generalization with fewer data samples. However, in the same way that FS does not allow for the full, higher-order control of FIS, graphical selection does not give the full, higher-order control of MIS.

**Higher Order Boltzmann** The 'fully-visible higher-order Boltzmann machine', extending 2-body correlations to all higher order interactions, was formulated long before it could be made practical (Sejnowski, 1986). Early works focus on binary variables with very small event spaces (Amari, 2001; Nakahara & Amari, 2002; Ganmor et al., 2011), mainly interested in biological applications like neuronal activity and protein interactions. These older works mainly ignored the computational issues associated with scaling to more serious sizes and accordingly remained limited in their application. The most closely related works to ours are the two works that have extended the sparse graphical modeling formulation to higher-order interactions, (Schmidt & Murphy, 2010) and (Min et al., 2014). Although not extending beyond binary variables, these works attempt scalability by formulating the hierarchical structure learning problem and trying to overcome the computational challenges of scaling higher-order Boltzmann machines to learning real-world data distributions.

Although these two works make strides towards defining the problem and attempting to scale beyond synthetic datasets, they still apply only to binary variables with many challenges remaining even for medium-scale datasets. Moreover, a deeper theoretical understanding of the statistical benefits had yet to be developed. We push further what is possible in practice by leveraging a theoretical grounding from tools in information geometry and efficient GPU-based training methods for modern applications. This progress, however, must be contrasted against the significant gap in capability when comparing to likelihood-free neural approaches.

**SOTA Generative Models** It is reminded that most recent work in distribution learning has shifted entirely away from having direct control over the distribution at all. Instead, enabling and later enabled by deep learning, recent models have been developed to instead reshape a large set of hidden or latent variables towards the distribution of the data, following the trail blazed by RBMs and DBMs (Hinton & Salakhutdinov, 2006; Salakhutdinov & Hinton, 2009) and extending into the modern day VAEs, GANs, and diffusion models (Kingma & Welling, 2014; Goodfellow et al., 2014; Ho et al., 2020).

In contrast to these methods, this work learns the hierarchical log-linear model on the data features, directly providing predictions of likelihood. For tabular datasets with interpretable variables, there is the great benefit of having interpretable insights into the data structure compared to what is available from latent approaches. Furthermore, it is imagined that this work's revisiting of the theoretical foundations for the statistical generalization properties of 'visible-only' generative modeling may ultimately have downstream effects on bettering our understanding of the generalization of generative modeling in more general cases including latent variables.

## 3 Refined Information

**Notation** Consider distributions over $d \in \mathbb{N}$ variable dimensions, with each feature $k \in [d] := \{1, \ldots, d\}$ having $I_k \in \mathbb{N}$ discrete (and disjoint) possibilities called events. We write indices as $i \in [I_1] \times \cdots \times [I_d]$ and index distributions as $p(i)$. When we deal with subsets of the features $S \subseteq [d]$ and their conjugates $-S := [d] - S$ (where we use $+$ and $-$ as shorthand for set union and set difference), we write the subindices $i_S \in I_S := \bigotimes_{k \in S} [I_k]$ and the marginal distributions $p^S(i_S) := \sum_{j_{-S} \in I_{-S}} p(i_S, j_{-S})$. We write the powerset as $\mathcal{P}([d]) := \{S \subseteq [d]\} \cong \{0, 1\}^d$ and a collection of interaction subsets as $\mathcal{I} \subseteq \mathcal{P}([d])$, or equally $\mathcal{I} \in \mathcal{P}(\mathcal{P}([d]))$.

To reiterate, we will write: $k \in [d], S \in \mathcal{P}([d]), \mathcal{I} \in \mathcal{P}(\mathcal{P}([d]))$, hence also $k \in S, S \in \mathcal{I}$.

Every distribution will be considered simultaneously as a discrete distribution and as a finite tensor, meaning that we interpret the tensor product on distributions as $(p^A \otimes p^B)(i_{A+B}) := p^A(i_A) \cdot p^B(i_B)$. We write $u$ to represent the uniform distribution and we will later write $p_{\text{trn}}$ and $p_{\text{val}}$ to represent the empirical training and validation distributions.

### 3.1 Information Theory

Information theory was born out of the fundamental contributions of Shannon (Shannon, 1948) defining the entropy of a variable and the mutual information (MI) between two variables. Shortly after, an extension to three or more variables was constructed with the multiple mutual information (MMI) (McGill, 1954). We

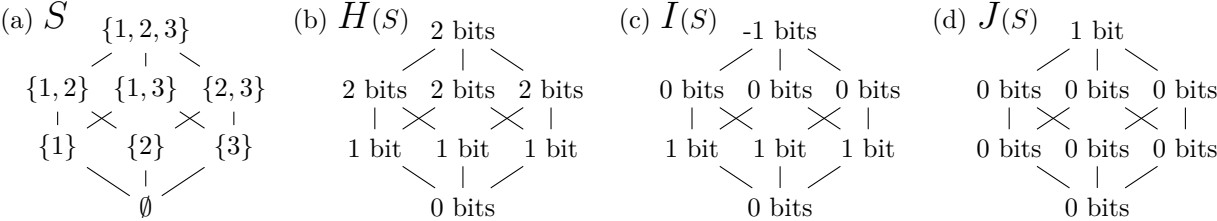

Figure 1: Three types of higher-order information ($H(p_S)$, $I(p_S)$, $J(p_S)$) defined for each $S \subseteq [3]$.

write the definitions of entropy, $H(p_S)$, and MI/MMI, $I(p_S)$, as:

$$H(p_S) := -\sum_{i_S} p_S(i_S) \log\{p_S(i_S)\}, \tag{1}$$

$$I(p_S) := \sum_{T \subseteq S} (-1)^{|T|-1} H(p_T), \tag{2}$$

$$J(p_S) := \sum_{T \subseteq S} (-1)^{|S|-|T|} D_{\mathrm{KL}}(p_T; u_T). \tag{3}$$

We additionally write the definition of $J(p_S)$, which is closely related to the original MI and MMI via $J(p_S) = (-1)^{|S|} I(p_S)$ for $|S| \geq 2$. As we will later see in the definitions to follow, this simple rephrasing is born out of our information geometric viewpoint rather than the original communication theory viewpoint.

In Figure 1, we can see their different properties on a simple distribution over three binary variables. In particular, take $X_1$ and $X_2$ to be Bernoulli and the third $X_3$ to be the XOR of the first two, $X_3 = X_1 \oplus X_2$. All three variables are symmetric and any one/ two of them is indistinguishable from a single/ pair of random bits. Accordingly, there is no mutual information between any of the pairs. However, despite one or two of the three variables seeming totally random, knowing all three variables simultaneously is completely different from the case of three independent variables. Accordingly, this distribution exemplifies the need to discuss 'purely third-order' types of information.

Unfortunately, $I(p_S)$ and $J(p_S)$ may return positive or negative values whenever $|S| \geq 3$, diminishing the ability to interpret MMI as 'information content' as is possible for the mutual information between two variables. This inspires our new definition in Equation 5 of Section 3.3 which generalizes mutual information but still returns a non-negative quantity for higher-order interaction information.

## 3.2 Information Geometry

It is fairly well known that we can model any discrete distribution by an exponential family:

$$q_\theta(i) := \exp\left\{ \sum_{S \subseteq [d]} \theta^S_{i_S} \right\}$$

for some continuous parameters $\theta^S \in \mathbb{R}^{I_S}$. It is lesser known that when equipped with KL divergence, this results in a Riemannian manifold called a statistical manifold. Due to the exponential structure, this further results in a dually flat or Bregman flat manifold, allowing for an even richer theory to be developed over the space (Rao, 1992; Amari & Nagaoka, 2000; Amari, 2016; Nielsen et al., 2017). The natural $\theta$ parameters and the dual expectation $\eta$ parameters $\eta^S_{i_S} := \mathbb{E}_{j \sim q_\theta(j)}[1(i_S = j_S)] = q^S_\theta(i_S)$ are 'orthogonal' to each other in a particular sense, obeying the Legendre transform and Bregman duality. Details in Appendix A.

Of particular importance for our work are the Pythagorean theorem and the projection theorem (Nagaoka & Amari, 1982), which imply the uniqueness of the distribution projected onto a flat submanifold of the distribution space as well as the convexity of the corresponding optimization problem. See Theorems 2 and 3 in Appendix A for details. In particular, for a fixed $p$ and a fixed collection of interactions $\mathcal{I} \subseteq \mathcal{P}([d])$, the

best forward $D_{\mathrm{KL}}$ approximation of $p$ within the submanifold $\mathcal{M}_{\mathcal{I}}$ allows for a unique projection $p_{\mathcal{I}}$:

$$\mathcal{M}_{\mathcal{I}} := \left\{ q_\theta : q_\theta(i) = \exp\left\{ \sum_{S \in \mathcal{I}} \theta^S_{i_S} \right\} \right\}, \qquad p_{\mathcal{I}} := \Pi_{\mathcal{M}_{\mathcal{I}}}(p) = \operatorname*{argmin}_{q \in \mathcal{M}_{\mathcal{I}}} \left\{ D_{\mathrm{KL}}(p;q) \right\}, \qquad (4)$$

where $\Pi_{\mathcal{M}_{\mathcal{I}}}$ denotes the projection onto that submanifold. In particular, the solution to our optimization problem is hence guaranteed to have a unique solution, and this solution respects the dually flat manifold.

### 3.3 Definition of Refined Information

In order to achieve our goal of defining a fine-grained and high-order definition of information content, we use this information geometry lens to define a sequence of projected distributions, and then use the distance (divergence) between each of the distributions in that sequence of projections as the definition of information content. Because mutual information can also be defined in this way, our definition gives a natural extension to higher-order and non-negative information content which we call **refined information**.

Let us say that a collection $\mathcal{I}$ is *hierarchical* if it is downwards closed with respect to subsets ($S \in \mathcal{I}$ and $T \subseteq S \Rightarrow T \in \mathcal{I}$). We can define a chain of collections such that:

$$\{\emptyset\} \subseteq \mathcal{I}_0 \subsetneq \mathcal{I}_1 \subsetneq \cdots \subsetneq \mathcal{I}_T \subseteq \mathcal{P}([d]).$$

We will say that a chain is *complete* whenever $\mathcal{I}_0 = \{\emptyset\}$ and $\mathcal{I}_T = \mathcal{P}([d])$ and is *hierarchical* whenever each $\mathcal{I}_t$ is hierarchical. We will hereafter restrict our attention to complete and hierarchical chains. Moreover, we will almost always considered *maximally-refined* chains, which is equivalent to saying $\mathcal{I}_t = (\mathcal{I}_{t-1} + S_t)$ for some subset $S_t$, for all choices of $t$. Any shorter chain can be extended into a maximal chain by further refining the sequence.

We use our chain of hierarchical collections in conjunction with the information geometry projection to construct in parallel a chain of distributions $p_{\mathcal{I}_0}, \ldots, p_{\mathcal{I}_T}$. These can be seen as the repeated projection onto submanifolds which slowly approach our target distribution: $\Pi_{\mathcal{I}_0}(p), \ldots, \Pi_{\mathcal{I}_T}(p)$. Accordingly, we know that each drop in divergence defines a unique and positive quantity which we will call the **refined information** from $\mathcal{I}$ to $\mathcal{J}$: $RI_{\mathcal{I} \to \mathcal{J}}(p) := D_{\mathrm{KL}}(p_{\mathcal{J}}; p_{\mathcal{I}}) = D_{\mathrm{KL}}(p; p_{\mathcal{I}}) - D_{\mathrm{KL}}(p; p_{\mathcal{J}})$. From this definition, it is clear that:

$$D_{\mathrm{KL}}(p;u) := \sum_{t=1}^{T} RI_{\mathcal{I}_{t-1} \to \mathcal{I}_t}(p),$$

where we recall that the uniform distribution $u$ is the null model in the space of finite distributions (all $\theta$ coordinates are zero).

**Definition 1.** The **refined information** of $S$ at $\mathcal{I}$ is:

$$RI_{\mathcal{I},S}(p) := RI_{\mathcal{I} \to (\mathcal{I}+S)}(p) = D_{\mathrm{KL}}(p_{\mathcal{I}+S}; p_{\mathcal{I}}). \qquad (5)$$

This leads to a full decomposition of the KL error as:

$$D_{\mathrm{KL}}(p;u) = \sum_{t=1}^{T} RI_{\mathcal{I}_{t-1}, S_t}(p). \qquad (6)$$

Accordingly, after fixing a chain, this formula attributes each positive drop in KL error to a single interaction set $S$. Since the goal of distribution learning is to reduce the KL divergence to zero, this decomposition allows for extremely fine-grained control by directly corresponding each effective parameter $\theta^S$ we may choose to include with a decrease in error. We discuss the implications of this for generalization performance in the coming Section 4.1.

## 4 MAHGenTa

Here we introduce the **M**ode-**A**ttributing **H**ierarchy for **Gen**erating **Ta**bular data (MAHGenTa) to tackle this doubly-exponential combinatorial problem and efficiently learn an arbitrary probability distribution.

Our procedure consists of two major components: (1) a mode interaction selection algorithm in conjunction with an early stopping procedure to guarantee a low gap between the train and test performances; and (2) an efficient gradient descent training algorithm which overcomes the challenges of the normalizing constant with energy-based modeling and a GPU-enabled pytorch implementation which extends existing Gibbs samplers to higher-order tensors.

$$\underset{\mathcal{I}, \theta_{\mathcal{I}}}{\operatorname{argmin}} \left( D_{KL}(p_{val}; \hat{q}_\theta^{\mathcal{I}}) \quad \text{where} \quad \hat{q}_\theta^{\mathcal{I}} = \underset{q_\theta \in \mathcal{M}_{\mathcal{I}}}{\operatorname{argmin}} \left( D_{KL}(p_{trn}; q_\theta^{\mathcal{I}}) \right) \right). \tag{7}$$

We write our learning objective as a bilevel optimization problem in Equation 7. Further details justifying this choice under the theoretical framework decomposing the KL error are provided in the Appendix.

## 4.1 Mode Interaction Selection

Even when ignoring the difficulties associated with learning the continuous $\theta$ parameters in Section 4.2, there are major practical challenges in finding a good collection $\mathcal{I}$ from the combinatorially explosive set of available choices. Particularly, the question of how to select a good collection of mode interactions which accurately describe the distribution without overfitting to the training set. We must leverage an appropriate heuristic for selecting interacting modes amongst the $2^d$ choices of interaction and $2^{2^d}$ choices of final collection.

We follow similar greedy heuristics as have been explored in previous literature based on the strong or weak 'heredity' assumption for choosing pairs (Peixoto, 1987; Bien et al., 2013). Generalizing this to higher order interactions allows us to only consider a polynomial number of candidate interactions. In particular, we will explore starting from the smallest $S$, only considering $S$ if its heredity score, $\omega(S)/|S|$, is greater than the threshold $\tau = 30\%$. Further discussion of heredity in Appendix B.4.

$$\omega(S) := |\{T : S = T \cup \{i\} \text{ for some } i\} \cap \{T : T \text{ has already been selected}\}| \tag{8}$$

Based on our theoretical developments in Equation (6), we would like to add each $\theta^S$ parameter which corresponds to the greatest amount of refined information, continuing until our validation KL error stops decreasing alongside our training KL error. Early stopping in this way is justified because every parameter of the log-linear model is an effective parameter, and sequential projections along the statistical manifold will cause our model to obey the classical underfitting-overfitting curve.

Unfortunately, exactly computing the refined information is difficult because for degree three and higher as there is no closed form available and one must resort to the continuous optimization approaches leveraged herein. Instead, we must a priori choose some heuristic measurement which corresponds with high refined information gain from including a specific mode interaction within the log-linear model. Accordingly, we use the absolute value of $J_S$ as introduced in Section 3.1 as an easy-to-compute alternative to $RI_S$. We present our search and learning algorithm in Algorithm 1 which continuously alternates between adding new mode interactions to the model and using gradient descent to train with the new parameters. Each subroutine is presented in full detail in the appendix.

## 4.2 Gradient Descent Learning

As mentioned in previous sections, the optimization of $\{\theta^S\}_{S \in \mathcal{I}}$ is always a convex problem. Accordingly, for a small enough learning rate, we can always guarantee the convergence of the gradient descent algorithm. Nevertheless, we find there are still multiple challenges to overcome for fast training of log-linear models when attempting to scale to real-world datasets.

We first recall the gradient of a log-linear model when optimizing for forward KL divergence is $\nabla_{\theta^S}[D_{\mathrm{KL}}(p_{\mathrm{trn}}; q_\theta)] = -\eta_{\mathrm{trn}}^S + \eta_\theta^S$ when evaluated on the empirical training distribution $p_{\mathrm{trn}}$. However, because the parameter space of the hierarchical model is invariant under constant shifts along any parameter tensor (so long as another parameter absorbs the negative shift), we will restrict each $\theta^S$ such that its sum across every mode/fiber is equal to zero. Practically, this leads to the use of the purified gradient:

$$\tilde{\nabla}_{\theta^S}[D_{\mathrm{KL}}(p_{\mathrm{trn}}; q_\theta)] = \sum_{T \subseteq S}(-1)^{|S|-|T|}(-\eta_{\mathrm{trn}}^T + \eta_\theta^T) \otimes u^{S-T}. \tag{9}$$

---

**Algorithm 1:** MAHGenTa Algorithm

---

1   MAHGENTA($\tau, \alpha, K, T$)
2     $Err_{\text{best}} \leftarrow \infty, \ \mathcal{I} \leftarrow \{\emptyset\}, \ \Theta \leftarrow \{0\}$
3     **while** $Error(\Theta) < Err_{\text{best}}$ **do**
4        $Err_{\text{best}} \leftarrow Error(\Theta)$
5        $\mathcal{J} \leftarrow$ NEXTAVAILABLEINTERACTIONS($\mathcal{I}, \tau$)
6        $\mathcal{K} \leftarrow$ TOPINTERACTIONS($\mathcal{J}, K$)
7        **for** $S \in \mathcal{K}$ **do**
8           $\theta^S \leftarrow \vec{0} \in \mathbb{R}^{I_S}$
9           $\Theta \leftarrow \Theta \cup \{\theta^S\}$
10        $\Theta \leftarrow$ GRADIENTDESCENT($\Theta, \alpha, T$)
11     **return** $\Theta$

---

To facilitate the modern applicability of our algorithm, we implement a GPU-based gradient descent training algorithm in Pytorch (Paszke et al., 2019). The major challenge of any gradient implementation for energy-based models is the calculation of the partition function or normalizing constant. Even after implementation tricks and virtualization of the tensor, exact computation is simply too slow to handle the billions of events which are possible in even medium-dimensional tabular datasets, and we must resort to a new variant of the classical Gibbs sampling approach (Geman & Geman, 1984; Gelfand, 2000) in conjunction with the annealed importance sampling technique (Neal, 2001). We find that the compound use of several tricks like this is critical for achieving the fastest implementation of gradient descent for higher-order energy based models, so we provide a detailed explanation of each trick in Appendix B.5.

## 5 Experiments

To first get a glimpse into the theoretical properties of refined information and the sample complexity of MAHGenTa we generate a suite of synthetic distributions by choosing random $\theta^S$ and sampling data from the induced distribution, details in the appendix. We demonstrate the impact of structure learning and the value of refined information in this setting where we have full control over the structure in the ground truth. We next apply our method to three real-world distributions from UCI machine learning datasets. The three datasets used are shown in Table 1 with their numbers of samples, features, and total possible events.

### 5.1 Synthetic Results

In Figure 2, we show the sample complexity of training when the underlying four-dimensional distribution has low complexity, medium complexity, or high complexity (top to bottom). For each of the three data distribution, we then train models of three different complexities and evaluate their train-time and test-time KL error. In the bottom row, we see how the underspecified, low-complexity model leads to underfitting which peaks at subpar performance. In the top row, we see how the overspecified, high-complexity model leads to overfitting which learns more slowly and less efficiently than the low-complexity model (even with multiple thousands of samples). In addition to showing the importance of matching the correct structure to achieve optimal performance, these experiments also show how achieving good generation performance automatically generalizes to classification performance.

In Figure 3, we plot all different values of refined information for our high complexity distribution. This gives some preliminary insights into the properties of refined information and the range of values a single interaction $S$ can take depending on the context $\mathcal{I}$. We additionally plot the 2D marginal refined information which corresponds to the classical definition of mutual information. Further discussion of its properties and applications to structure learning and generalization are saved for Appendices B and C.

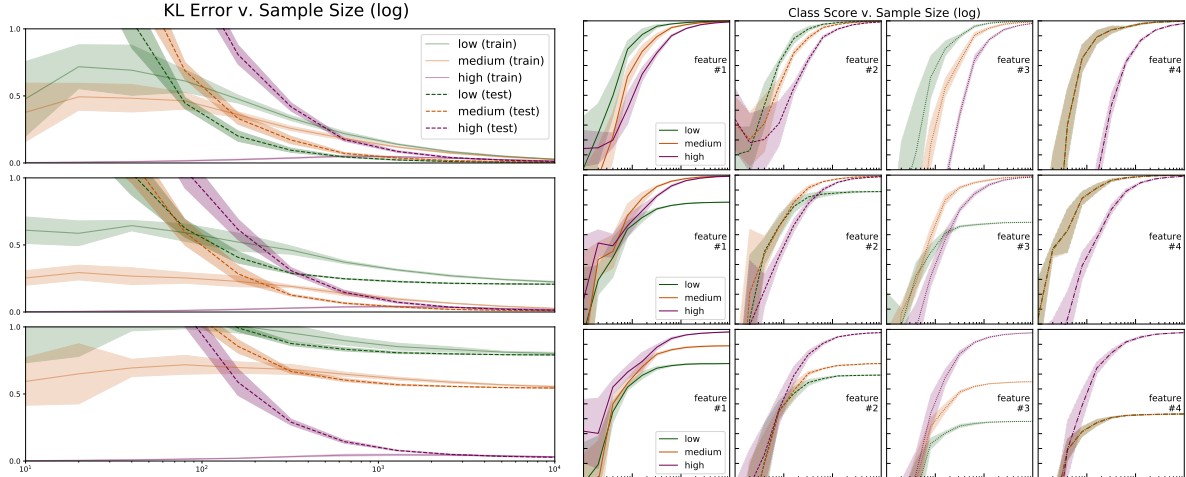

Figure 2: Model Performance vs. Number of Training Samples. Left three panels: KL error as a function of sample size. Right twelve panels: Normalized classification error as a function of sample size. Performance is evaluated across three different model complexities. Each row corresponds to the complexity of the *underlying data distribution* which from top to bottom has low complexity, medium complexity, and high complexity. The top row shows the high complexity model overfitting and slowly fitting. The bottom row shows the low complexity model underfitting. Error bars are with respect to 5 different resampling of the synthetic training dataset.

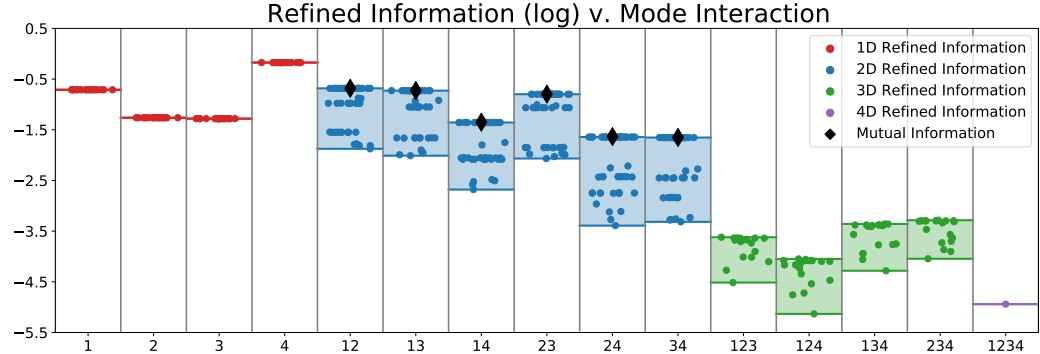

Figure 3: Synthetic four-dimensional data with all possible values of refined information plotted. Each column represents an interaction $S \subseteq [d]$. Each point represents a particular value of $RI_{\mathcal{I},S}$ for some possible pair $(\mathcal{I}, S)$. Random horizontal spread is used for visual clarity.

## 5.2 Real-World Results

For the real-world datasets, we first demonstrate the tuning of our MIS hyperparameters (heredity strength, heuristic norm, and loss function) on a small subset of the real-world dataset. By using only the first ten dimensions of the mushroom dataset, we work in a regime where the exact gradients can be readily calculated and we perform our algorithm with collections of up to size 300.

Figure 4 shows the performance in terms of the capacity curves, plotting the training and validation errors as a function of the size of the interaction collection, which is a measure of the log-linear model's capacity. We train significantly beyond the point of early stopping to help fully illustrate the underfitting-overfitting behavior of the log-linear model. This provides empirical support for our theoretically principled approach of early stopping as soon as the validation error stops improving alongside the training error. Further hyperparameters are provided in the appendix. Overall, we find that using the weak hierarchy of 30%

Table 1: Statistics for real-world datasets.

|  | $n$ | $d$ | $|I_{[d]}|$ |
|---|---|---|---|
| **mushroom** | 8,124 | 23 | $2.4e14$ |
| **adults** | 32,561 | 14 | $6.5e11$ |
| **breast cancer** | 286 | 10 | $6.0e05$ |

strength was the most effective choice for achieving minimal validation error for our MAHGenTa algorithm and we keep this choice consistent throughout.

We then apply these hyperparameters to our two large-scale datasets where we cannot directly calculate the KL divergence and resort to the AIS approximation discussed in Section 4.2. For our third real-world dataset, exact KL gradients are still calculable with an event space smaller than one million. We compare against a Boltzmann machine and an independent distribution also trained with gradient descent on the same objective. In Table 2, we compare our approach which has the capacity to learn sparse and higher-order structures against both the 1-body and 2-body log-linear models. We find that our MAHGenTa approach is able to consistently deliver improvements in generation performance in terms of the KL divergence or log-likelihood.

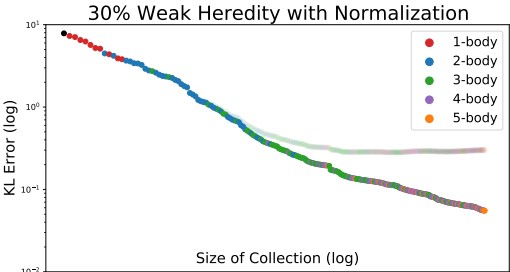 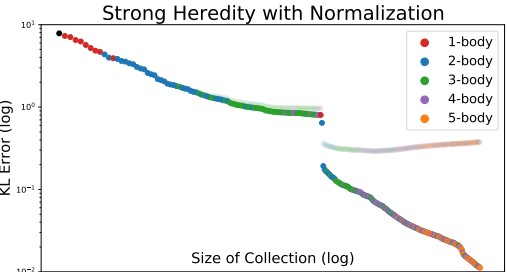

Figure 4: Comparison of the training (dark) and validation (light) error as we continue to add interaction subsets to our collection. On the log-log plot, we see clear distinction of the underfitting phase and the overfitting phase. We also see how assuming strong heredity can lead to 'discontinuities' in the performance, caused by important higher-order interactions being blocked in the greedy algorithm.

Table 2: KL divergence across all real-world datasets.

|  | mushroom | adults | breast cancer |
|---|---|---|---|
| independent (1D) | $15.477 \pm 0.056$ | $8.692 \pm 0.047$ | $5.991 \pm 0.210$ |
| boltzmann (2D) | $4.472 \pm 0.069$ | $6.444 \pm 0.042$ | $5.652 \pm 0.105$ |
| mahgenta (3D+) | $\mathbf{2.212 \pm 0.062}$ | $\mathbf{5.832 \pm 0.012}$ | $\mathbf{5.176 \pm 0.052}$ |

### 5.3 Discussion

In Table 3, we see how the training of a generative model automatically leads to emergent capabilties in classification via the mode interactions simplifying into feature interactions. In particular, the energy-based MAHGenTa and Boltzmann machine are able to simultaneously predict across multiple classes, unlike the discriminative approaches which must be retrained for each task. Although the discriminative approaches have the advantage of reusing the dataset to learn only one of the conditional distributions at a time, the generative approaches nevertheless yield a comparable accuracy performance across a variety of tasks simultaneously.

In the adults dataset, we can clearly see how a single generative model trained to adequately model the data easily obtains good accuracy not only for the original target of income level, but also sensitive features

Table 3: Class-wise accuracy across all real-world datasets.

| | mushroom | | | adults | | | breast cancer | |
|---|---|---|---|---|---|---|---|---|
| | poison | odor | habitat | income | race | gender | recurrence | malignance |
| mahgenta | 99.7 | 79.7 | **66.0** | 85.2 | 86.5 | **84.5** | **80.2** | **51.6** |
| boltzmann | 98.2 | 78.5 | 63.4 | 84.2 | 84.9 | 83.6 | 72.1 | 50.4 |
| logistic regression | **100.0** | **80.6** | 65.8 | **85.6** | **88.0** | 84.4 | 71.3 | 42.7 |
| naive bayes | 94.8 | 78.6 | 63.3 | 81.6 | 85.3 | 82.1 | 72.0 | 44.1 |

like race and gender. In the classification setting, it may be unclear that a model is biased using sensitive features to predict income; however, in our energy-based model working directly on the observed variables, the learned connections between variables are made explicit. This could have implications for algorithm fairness approaches, where removing sensitive feature labels from the training data is not sufficient to remove the fundamental bias which exists within the dataset. In contrast, biased energy terms in the log-linear model could be directly inspected, analyzed, and removed.

## 6 Conclusion

Overall, we find that refined information opens up many directions for further exploration of higher-order information and that mode-interaction-selection for hierarchical log-linear modeling is an effective tool in reducing the number of parameters to be learned in a principled way. Theoretical developments allow for a complete decomposition of the KL error in terms of the refined information content. The regularizing effect of choosing simpler structure is made clear on both synthetic and real-world datasets, with an easy-to-use early stopping heuristic to achieve optimal performance. The benefits of generative distribution learning as a general pretraining objective for multiple downstream tasks are also reinforced.

## Acknowledgements

This work was supported by JST, CREST Grant Number JPMJCR22D3, Japan.

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

# A Information Geometry

## A.1 Necessary Background

Information geometry is the field which treats the set of parameters $\theta \in \mathbb{R}^p$ as a manifold with a Riemannian metric, called the statistical manifold. In our case, we will focus on the structure introduced by KL divergence. Accordingly, we will write the divergence as forward KL divergence $D(p;q) = D_{\mathrm{KL}}(p;q)$, the dual divergence as reverse KL divergence $D^*(p;q) = D_{\mathrm{KL}}(q;p)$, the Bregman potential as negative entropy $\varphi(p_\theta) = -\sum_i p_\theta(i) \log p_\theta(i)$, and the dual Bregman potential as free energy $\psi(p_\theta) = \log\{\sum_i \exp\{\sum_j \theta_j e_j(i)\}\}$. For now, we write $e_j$ as the one hot basis for the discrete space $e_j(i) := 1(i = j)$, further discussion on alternate coordinate systems will be had in the next subsection.

We write the distribution corresponding to some $\theta$ parametrization as:

$$p_\theta(i) = \exp\left\{\sum_j \theta_j e_j(i) - \psi(\theta)\right\} \tag{10}$$

where it can be seen that $\psi(p_\theta)$ is already chosen such that

$$\sum_i p_\theta(i) = \sum_i \left\{\exp\left\{\sum_j \theta_j e_j(i)\right\} \cdot \exp\left\{-\log\sum_k \exp\sum_l \theta_l e_l(k)\right\}\right\}$$

$$= \left(\sum_i \exp\sum_j \theta_j e_j(i)\right) \cdot \left(\sum_k \exp\sum_l \theta_l e_l(k)\right)^{-1} = 1.0 \tag{11}$$

Using the one hot basis $e_j$, this means that we may write $p_\theta(i) = \exp\{\theta_i\}/(\sum_j \exp\{\theta_j\})$.

However, to ensure there are no redundancies in the manifold, we are required to choose to drop one coordinate. Otherwise, we could rescale by shifting by a constant. It is convention to drop the first coordinate, so $p_\theta(1) = 1/(\sum_j \exp\{\theta_j\})$. Again, we discuss our alternative later in Section A.3.

This choice of $D, D^*, \varphi$, and $\psi$ allows for a duality between the $\theta$ coordinates and the $\eta$ coordinates corresponding to the expectations of the basis functions $\eta_i = \mathbb{E}_{p_\theta}[e_i]$, which has the following nice consequences:

**Theorem 1.** (**Legendre Transform Formula**, Theorem 1.1 of (Amari & Nagaoka, 2000))

$$D_{\mathrm{KL}}(p;q) = \varphi(p) + \psi(q) - \eta(p) \cdot \theta(q) \tag{12}$$

*Proof.* $D_{\mathrm{KL}}$ is a Bregman divergence of the form $D_\varphi$, meaning that $D_{\mathrm{KL}}(p;q) = \varphi(p) - \varphi(q) - \nabla\varphi(q) \cdot (p - q)$. Because we know from the Bregman dually flat structure that $\theta = \eta^* = \nabla\varphi(\eta)$ and $\eta = \theta^* = \nabla\psi(\theta)$ and we also have the Legendre duality by our choice of $\varphi$ and $\psi$, giving $\varphi(q) + \psi(q^*) = \eta(q) \cdot \theta(q)$, we have that:

$$D_{\mathrm{KL}}(p;q) = \varphi(p) - \varphi(q) - \theta(q) \cdot (\eta(p) - \eta(q)) \tag{13}$$

$$= \varphi(p) + [\psi(q) - \eta(q) \cdot \theta(q)] - \theta(q) \cdot (\eta(p) - \eta(q)) \tag{14}$$

$$= \varphi(p) + \psi(q) - \theta(q) \cdot \eta(p) \tag{15}$$

$\square$

**Theorem 2.** (**Pythagorean Theorem**, Theorem 1.2 of (Amari & Nagaoka, 2000)) Given a dually flat manifold, with distribution $p, q, r$ such that $p$ and $q$ are connected by an $\eta$ geodesic and $q$ and $r$ are connected by a $\theta$ geodesic, which are orthogonal, then the generalized Pythagorean theorem holds:

$$D_{\mathrm{KL}}(p;r) = D_{\mathrm{KL}}(p;q) + D_{\mathrm{KL}}(q;r). \tag{16}$$

*Proof.* We may write that:

$$D(p;q) + D(q;r) = [\varphi(p) + \psi(q) - \eta(p) \cdot \theta(q)] + [\varphi(q) + \psi(r) - \eta(q) \cdot \theta(r)] \tag{17}$$
$$= [\varphi(p) + \psi(r) - \eta(p) \cdot \theta(r)] + [\eta(p) \cdot \theta(r) + [\varphi(q) + \psi(q)] - \eta(p) \cdot \theta(q) - \eta(q) \cdot \theta(r)] \tag{18}$$
$$= D(p;r) - [\eta(p) - \eta(q)] \cdot [\theta(q) - \theta(r)] \tag{19}$$
$$= D(p;r) \tag{20}$$

where the last step follows by the orthogonality of the geodesics. For each coordinate $i$, we have that either $[\theta(p) - \theta(q)]_i$ is zero or $[\eta(q) - \eta(r)]_i$ is zero, meaning their dot product is zero overall. $\qquad\square$

**Theorem 3.** (**Projection Theorem**, Theorem 1.4 of (Amari & Nagaoka, 2000)) Given a dually flat manifold and considering a submanifold $\mathcal{M}$, for example one defined by a subset of the coordinates, the point that minimizes the divergence is the dual geodesic projection and the point that minimizes the dual divergence is the geodesic projection.

*Proof.* Proof is completed by considering the geodesic project and dual geodesic projections in a small neighborhood where the Pythaogrean theorem then shows that any small deviation can only create a positive increase in divergence or dual divergence (due to the non-negativity of divergences). This confirms that the dual projection and projection are indeed a critical point. $\qquad\square$

## A.2 Hierarchical Coordinates

In this work, we follow the hierarchical coordinate scheme of (Ghalamkari et al., 2023) in order to adequately respect the many-body structure of the mode interactions between the different variables.

$$q_\theta(i) := \exp\left\{\sum_{S \subseteq [d]} \theta_{i_S}^S\right\}$$

It should now be understood how naively using these coordinates results in an overspecified number of dimensions compared to the actual manifold (which is $(I-1)$-dimensional). Accordingly, we first consider the approach of (Ghalamkari et al., 2023) which zeroes out the first coordinates at each hierarchical level. See the later examples in Section A.3.

These hierarchical coordinates are already sufficient for defining refined information, so we first confirm this hierarchical representation remains valid under information geometry.

First writing the free energy constant $\theta^\emptyset = \psi(\theta)$ as:

$$\theta^\emptyset = -\log\left\{\sum_i \exp\left\{\sum_{\emptyset \subsetneq S \subseteq [d]} \theta_{i_S}^S\right\}\right\}.$$

As mentioned, we can envision the parameter $\theta^\emptyset$ either as a function of the other $\theta^S$ or as a parameter which is then constrained by the other parameters. Although the function perpsective is needed to nicely align with the theory of information geometry, we also find it useful to take the latter perspective during the practical fitting of these energy-based model. For simplicity, we will for now take the former perspective and look at the derivative of $\theta^\emptyset$ or the free energy taken as a function of the other parameters (but this can

also be thought of as the derivative of the constraining equation).

$$\frac{\partial}{\partial \theta_{j_T}^T}\left\{\theta^\emptyset\right\} = -\frac{\partial}{\partial \theta_{j_T}^T}\log\left\{\sum_i \exp\left\{\sum_{\emptyset\subsetneq S\subseteq[d]}\theta_{i_S}^S\right\}\right\}$$

$$= -\left\{\sum_i \exp\left\{\sum_{\emptyset\subsetneq S\subseteq[d]}\theta_{i_S}^S\right\}\right\}^{-1}\cdot\frac{\partial}{\partial \theta_{j_T}^T}\left\{\sum_i \exp\left\{\sum_{\emptyset\subsetneq S\subseteq[d]}\theta_{i_S}^S\right\}\right\}$$

$$= -\left\{\exp\left\{-\theta^\emptyset\right\}\right\}^{-1}\cdot\left\{\sum_i \frac{\partial}{\partial \theta_{j_T}^T}\exp\left\{\sum_{\emptyset\subsetneq S\subseteq[d]}\theta_{i_S}^S\right\}\right\}$$

$$= -\exp\left\{\theta^\emptyset\right\}\cdot\left\{\sum_i \exp\left\{\sum_{\emptyset\subsetneq S\subseteq[d]}\theta_{i_S}^S\right\}\cdot\delta_{i_T,j_T}\right\}$$

$$= -\left\{\sum_i \delta_{i_T,j_T}\cdot\exp\left\{\sum_{S\subseteq[d]}\theta_{i_S}^S\right\}\right\}$$

$$= -\sum_i \delta_{i_T,j_T}\cdot q_\theta(i) = -\sum_i \delta_{i_T,j_T}\cdot\eta_i^{[d]} = -\sum_{i_T}\sum_{i_{-T}}\delta_{i_T,j_T}\cdot\eta_i^{[d]} = -\sum_{i_T}\delta_{i_T,j_T}\cdot\eta_{i_T}^T = -\eta_{j_T}^T.$$

Now it will be easy for us to take the derivative of the objective function, the forward KL-divergence:

$$D_{KL}(p_{trn};q_\theta) = \sum_i p_{trn}(i)\cdot\log\left(\frac{p_{trn}(i)}{q_\theta(i)}\right).$$

Let us write:

$$\frac{\partial}{\partial \theta_{j_T}^T}D_{KL}(p_{trn};q_\theta) = \frac{\partial}{\partial \theta_{j_T}^T}\sum_i p_{trn}(i)\log(p_{trn}(i)) - p_{trn}(i)\log(q_\theta(i))$$

$$= 0 - \frac{\partial}{\partial \theta_{j_T}^T}\sum_i p(i)\log(q(i)) = -\sum_i \frac{\partial}{\partial \theta_{j_T}^T}p^{trn}(i)\log(q^\theta(i))$$

$$= -\sum_i p^{trn}(i)\cdot\frac{\partial}{\partial \theta_{j_T}^T}\log(q^\theta(i)) = -\sum_i p^{trn}(i)\cdot\frac{\partial}{\partial \theta_{j_T}^T}\left\{\sum_{S\subseteq[d]}\theta_{i_S}^S\right\}$$

$$= -\sum_i p^{trn}(i)\cdot\frac{\partial}{\partial \theta_{j_T}^T}\left\{\theta^\emptyset + \theta_{i_T}^T\right\} = -\sum_i p^{trn}(i)\cdot\frac{\partial}{\partial \theta_{j_T}^T}\left\{\theta^\emptyset\right\} - \sum_i p^{trn}(i)\cdot\frac{\partial}{\partial \theta_{j_T}^T}\left\{\theta_{i_T}^T\right\}$$

$$= -\frac{\partial}{\partial \theta_{j_T}^T}\left\{\theta^\emptyset\right\} - \sum_i p^{trn}(i)\cdot\delta_{i_T,j_T}$$

$$= -\left\{-\eta_{j_T}^{\theta,T}\right\} - p^{trn,T}(j_T) = \eta_{j_T}^{\theta,T} - \eta_{j_T}^{trn,T}.$$

This completes the calculation of the gradient for the parametrization of the probability distribution. We reiterate that this is the derivative with the free energy constraint, but without any additional constraints to enable identifiability of the parameters. In particular, this means that multiple $\theta$ parameters can correspond to the same probability distribution. This is alleviated by the zeroing out to restrict the manifold as done in (Ghalamkari et al., 2023). In the next section, we introduce our final version of the information-geometric coordinates, which instead use $\theta$ coordinates which are both hierarchical and centered, which we find to be critical for the practical deployment of these methods.

## A.3 Centered and Hierarchical Coordinates

In this section, we first revisit the zeroing out constraints of Sections A.1 and A.2 using explicit examples before introducing the centered coordinates which we introduce as an alternative with computationally nicer properties needed for practical implementation.

**Textbook Constraints**   The textbook method of providing additional constraints is to zero out all of the unnecessary parameters for the probability distribution. This is typically done by zeroing out the 'corners'

of each $\theta^S$ parameter via:

$$\theta^S_{i_S} = 0 \qquad \text{if } i_s = 1 \text{ for any } s \in S.$$

We omit the exact details of showing this will represent each probability distribution with a unique set of parameters; however, we note that this leaves each $\theta^S$ with $\prod_{s \in S}(|I_s| - 1)$ parameters out of the original $\prod_{s \in S}(|I_s|) = |I_S|$. Accordingly, all of the $\theta^S$ parameters together have $|I_{[d]}|$ degrees of freedom, and after the sum-to-one constraint from the last section on $\theta^\emptyset$, there are the required $|I_{[d]}| - 1$ degrees of freedom.

In the textbook formulation, these zeroed out coefficients are usually completely dropped from consideration. After additionally considering the $\theta^\emptyset$ as a function of the other parameters rather than another parameter itself, we are left with a parametrization of the manifold which has the same number of parameters are there are intrinsic dimensions in the manifold.

We could come to an equivalent such parametrization with any set of choices for the 'base events' $i'_s \in [I_s]$ instead of simply choosing the first event. However, some choices can be arbitrarily worse than others and searching over all possible choices of base events is much more work than it is worth. We will briefly introduce the concerns of numerical stability in some small, low-dimensional distributions. On a one-dimensional distribution with four outcomes, this would become:

$$p(1) = e^{\theta^\emptyset}, \qquad p(2) = e^{\theta^\emptyset + \theta^1_2}, \qquad p(3) = e^{\theta^\emptyset + \theta^1_3}, \qquad p(4) = e^{\theta^\emptyset + \theta^1_4}.$$

In this case, we can write the closed form solution as:

$$\theta^\emptyset = \log(p(1)), \qquad \theta^1_2 = \log(\frac{p(2)}{p(1)}), \qquad \theta^1_3 = \log(\frac{p(3)}{p(1)}), \qquad \theta^1_4 = \log(\frac{p(4)}{p(1)}).$$

Fortunately, this seems to mean that as long as $p(1)$ is not extremely large or extremely small compared to the other probabilities, there will not be a huge amount of issues with our arbitrary choice of base event for the exponential distribution. However, the risks associated with this arbitrary choice will only continue to grow and compound as we increase the dimensionality and the number of events.

In two dimensions, we can see:

$$p(1,1) = e^{\theta^\emptyset}, \qquad p(1,2) = e^{\theta^\emptyset + \theta^2_2}, \qquad p(1,3) = e^{\theta^\emptyset + \theta^2_3},$$
$$p(2,1) = e^{\theta^\emptyset + \theta^1_2}, \qquad p(2,2) = e^{\theta^\emptyset + \theta^1_2 + \theta^1_2 + \theta^{12}_{22}}, \qquad p(2,3) = e^{\theta^\emptyset + \theta^1_2 + \theta^1_3 + \theta^{12}_{23}}.$$

Again, we may write a closed form solution as:

$$\theta^\emptyset = \log(p(1,1)), \qquad \theta^2_2 = \log\left(\frac{p(1,2)}{p(1,1)}\right), \qquad \theta^2_3 = \log\left(\frac{p(1,3)}{p(1,1)}\right),$$
$$\theta^2_2 = \log\left(\frac{p(1,2)}{p(1,1)}\right), \qquad \theta^{12}_{22} = \log\left(\frac{p(2,2) \cdot p(1,1)}{p(1,2) \cdot p(2,1)}\right), \qquad \theta^{12}_{23} = \log\left(\frac{p(2,3) \cdot p(1,1)}{p(1,3) \cdot p(2,1)}\right).$$

It can be imagined how increasingly high dimensional distributions would only exacerbate the issues of potentially imbalanced events within the space, especially the high dependence on the probability $p(1, \ldots, 1)$. It is important to also remember that for our main application, there does not exist a closed form solution for the $\theta$ parameters and we must use gradient-based learning approaches, necessitating an adequate handling of these potential numerical issues.

**Balanced Parametrization** Given these concerns of significant and unnecessary challenges in the gradient-based learning process for the $\theta$ parameters, we instead leverage an alternative identifiability constraint which is more compatible with initialization at the all zeroes vector. In particular, we balance the $\theta^S$ tensor around zero by assuming that each fiber sums to zero:

$$\sum_{j_s} \theta^S_{i_{(S-s)}, j_s} = 0 \qquad \text{for any } s \in S \text{ for any } i_{(S-s)} \in I_{(S-s)}. \tag{21}$$

It is again relatively straightforward to verify that this gives a unique parametrization for every probability distribution in the manifold. We can also see that this condition can be written in a more tensorial form as:

$$\sum_{j_s} \theta_{j_s}^S = \vec{0} \qquad \text{for any } s \in S,$$

where $\vec{0}$ refers to the zero tensor $\vec{0} \in \mathbb{R}^{I_{S-s}}$. Let us also recall the $\eta$ parameters defined as $\eta_{i_S}^S := \mathbb{E}_{j \sim q(j)}[1(i_S = j_S)] = q^S(i_S) = \sum_{j_{-S}} q(i_S, j_{-S})$. We may immediately see that the $\eta$ parameters automatically obey a similar tensorial constraint as:

$$\sum_{j_s} \eta_{j_s}^S = \eta^{S-s} \qquad \text{for any } s \in S.$$

Both the $\theta$ and the $\eta$ sets of parameters form a tower structure made of tensors. We may immediately try to purify the $\eta$ tower in a similar way to our $\theta$'s via the principle of inclusion-exclusion (related to the mobius inversion of the zeta function), namely let us write:

$$\pi^S := \sum_{T \subseteq S} (-1)^{|S|-|T|} \cdot \eta^T \otimes u^{S-T}. \tag{22}$$

It is fairly straightforward to see that these new $\pi$ parameters obey the centralized condition:

$$\sum_{j_s} \pi_{j_s}^S = \vec{0} \qquad \text{for any } s \in S$$

and that moreover the $\eta$ variables are recoverable in the opposite direction via:

$$\eta^S := \sum_{T \subseteq S} \pi^T \otimes u^{S-T}.$$

More importantly, we may write the purified gradient of Equation 9 in terms of $\pi$ which further corresponds to the constrained gradient of the KL divergence:

$$\tilde{\nabla}_{\theta^S} \left[ D_{\mathrm{KL}}(p_{\mathrm{trn}}; q_\theta) \right] = -\pi_{\mathrm{trn}}^S + \pi_\theta^S.$$

Although this is already sufficient for the purposes of our work, but let us proceed slightly to round out the discussions about our parametrization as it relates to the typical topics of information geometry. First, we recall that the typical Bregman divergences are forward and backwards KL with Bregman functions of free energy and total entropy. We have already computed the $\theta$ derivatives for free energy when looking at the derivative of forward KL, but let us also take a look at the backwards case with entropy.

First recall that

$$H(p) = -\sum_i p(i) \log \left( p(i) \right)$$

and also that

$$\frac{\partial}{\partial x} f \log(f) = [f \cdot 1/f + 1 \cdot \log(f)] \cdot f' = [1 + \log(f)]f',$$

$$-\frac{\partial}{\partial \pi_{j_T}^T} H(q_\theta) = \frac{\partial}{\partial \pi_{j_T}^T} \sum_i \left( \sum_S \pi_{i_S}^S \otimes u_{i_{-S}}^{-S} \right) \log \left( \sum_S \pi_{i_S}^S \otimes u_{i_{-S}}^{-S} \right)$$

$$= \sum_i \left[ 1 + \log \left( \sum_S \pi_{i_S}^S \otimes u_{i_{-S}}^{-S} \right) \right] \cdot \frac{\partial}{\partial \pi_{j_T}^T} \left( \sum_S \pi_{i_S}^S \otimes u_{i_{-S}}^{-S} \right)$$

$$= \sum_i \left[ 1 + \log \left( q_\theta(i) \right) \right] \cdot \left( \sum_S \frac{\partial}{\partial \pi_{j_T}^T} \pi_{i_S}^S \otimes u_{i_{-S}}^{-S} \right)$$

$$= \sum_i \left[ 1 + \sum_{S \subseteq [d]} \theta_{i_S}^S \right] \cdot \left( \delta_{i_T, j_T} \cdot u_{i_{-T}}^{-T} \right)$$

$$= \sum_{i_{-T}} \sum_{i_T} \delta_{i_T, j_T} \cdot \left[ 1 + \sum_{S \subseteq [d]} \theta_{i_S}^S \right] \cdot \frac{1}{|I_T|}$$

$$= \sum_{i_{-T}} \left[ 1 + \sum_{S \subseteq [d]} \theta_{j_T \cap S, i_{S-T}}^S \right] \cdot \frac{1}{|I_T|}$$

$$= \frac{1}{|I_T|} \sum_{i_{-T}} \left[ 1 + \sum_{S \subseteq T} \theta_{j_{T \cap S}}^S + \sum_{S \not\subseteq T} \theta_{j_{T \cap S}, i_{S-T}}^S \right]$$

$$= \left[ 1 + \sum_{S \subseteq T} \theta_{j_{T \cap S}}^S \right] + \left[ \sum_{S \not\subseteq T} \frac{1}{|I_T|} \sum_{i_{-T}} \theta_{j_{T \cap S}, i_{S-T}}^S \right]$$

$$= \left[ 1 + \sum_{S \subseteq T} \theta_{j_S}^S \right].$$

Continuing on,

$$-\frac{\partial}{\partial \pi_{j_T}^T} D_{KL}(q_\theta; p_{trn}) = -\frac{\partial}{\partial \pi_{j_T}^T} H(q_\theta) - \frac{\partial}{\partial \pi_{j_T}^T} \sum_i \left( \sum_S \pi_{i_S}^S \otimes u_{i_{-S}}^{-S} \right) \log \left( p_{trn}(i) \right)$$

$$= \left[ 1 + \sum_{S \subseteq T} \theta_{j_S}^S \right] - \sum_i \left( \delta_{i_T, j_T} \cdot u_{i_{-T}}^{-T} \cdot \sum_{S \subseteq [d]} \theta_{i_S}^{trn, S} \right)$$

$$= \left[ 1 + \sum_{S \subseteq T} \theta_{j_S}^S \right] - \left[ 1 + \sum_{S \subseteq T} \theta_{j_S}^{trn, S} \right].$$

Using again the purification of the tower of $\theta$'s leaves us only with the top of the sum of $\theta$'s, meaning that the purified derivative of the free energy is only left with the corresponding $\theta_{j_T}^T$ parameter. Finally, we can write:

$$\tilde{\nabla}_{\pi^S} \left[ D_{KL}(q_\theta; p_{trn}) \right] = \theta^S - \theta_{trn}^S.$$

# B Experimental Details

## B.1 Additional Results

In Figures 5 and 6, we show the capacity curves across all sets of MIS hyperparameters chosen for the experiments with the 10-dimensional subset of the 23-dimensional mushroom datasets.

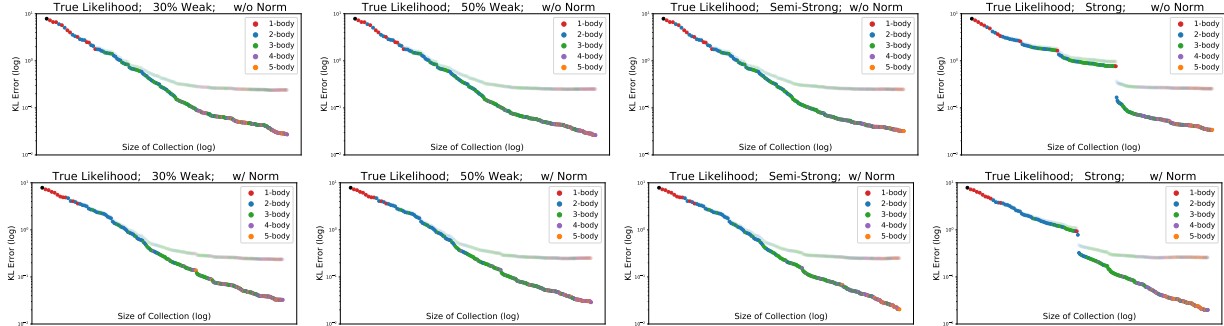

Figure 5: All hyperparameters of heredity strength and parameter count renormalization. Top 8: Full likelihood training.

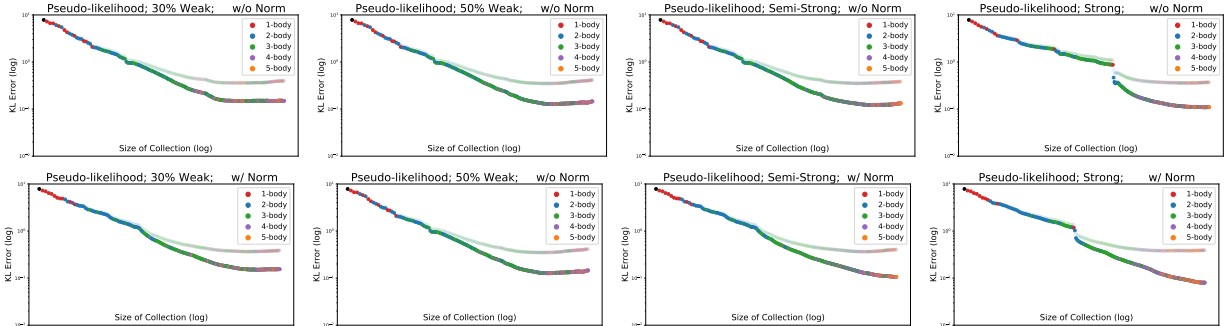

Figure 6: All hyperparameters of heredity strength and parameter count renormalization. Bottom 8: Pseudolikelihood training (masking turned off to avoid $-\infty$).

Table 4: Best validation KL error from all MIS hyperparameters and all many-body solutions.

| | | Heredity | | | |
|---|---|---|---|---|---|
| **Likelihood Type** | **Param Count Renorm** | 30% Weak | 50% Weak | Semi | 100% Strong |
| True Likelihood | with | **0.2359** | 0.2466 | 0.2445 | 0.2559 |
| True Likelihood | without | 0.2372 | 0.2489 | 0.2472 | 0.2555 |
| Pseudolikelihood | with | 0.3659 | 0.3533 | 0.3659 | 0.3817 |
| Pseudolikelihood | without | 0.3589 | 0.3498 | 0.3519 | 0.3583 |

| | **Many-Body** (no sparsity) | | |
|---|---|---|---|
| **Likelihood Type** | 1D | 2D | 3D |
| True Likelihood | 4.6062 | 0.8281 | 0.2644 |
| Pseudolikelihood | 4.6062 | 1.5005 | 0.6579 |

The search over these hyperparameters can be seen as a comparison to previous works (Schmidt & Murphy, 2010; Min et al., 2014) because of their use of other types of stronger hierarchical assumptions. Moreover, the stagewise-selection procedures can also be seen as a special case of this mode interaction selection framework. Accordingly, the only missing component of a full comparison to these previous works would be tuning over

the L1 regularization parameter. We find unlike those works, tuning an L1 parameter is not as necessary in our work due to our L0 selection with the MIS algorithm and our theoretically supported early-stopping procedure. It is moreover emphasized that both previous works have no available code and regardless were only designed for binary variables, making them inapplicable to any of our datasets used herein.

## B.2 Synthetic Datasets

We generate small synthetic distributions by first drawing $\theta^S$ from a Gaussian distribution with unit covariance. We then center each $\theta^S$ such that the sum across any mode is equal to zero. Afterwards, if there are any $S$ we have not yet zeroed out from the model, then we do this now. Finally, we compute the probability distribution as the exponential of the sum of the $\theta$ parameters and compute the renormalization constant if necessary. Synthetic training datasets are then drawn iid from this final distribution.

In our experiments, we use $d = 4$ dimensional distributions inside $I_{[d]} = [5] \times [5] \times [5] \times [5]$. For the low, medium, and high complexity distributions, we use the following sparsity patterns:

$$\begin{aligned} \text{low complexity} \quad & \mathcal{I}^*_{\text{low}} = \{\emptyset, 1, 2, 3, 4, 12, 14, 23\} \\ \text{medium complexity} \quad & \mathcal{I}^*_{\text{med}} = \{\emptyset, 1, 2, 3, 4, 12, 13, 14, 23, 123\} \\ \text{high complexity} \quad & \mathcal{I}^*_{\text{high}} = \{\emptyset, 1, 2, 3, 4, 12, 13, 14, 23, 24, 34, 123, 124, 134, 234, 1234\} \end{aligned}$$

We use sample sizes of $[10,20,40,80,160,320,640,1280,2560,5120,10240]$ and plot from 10 to 10,000 on a logarithmic scale.

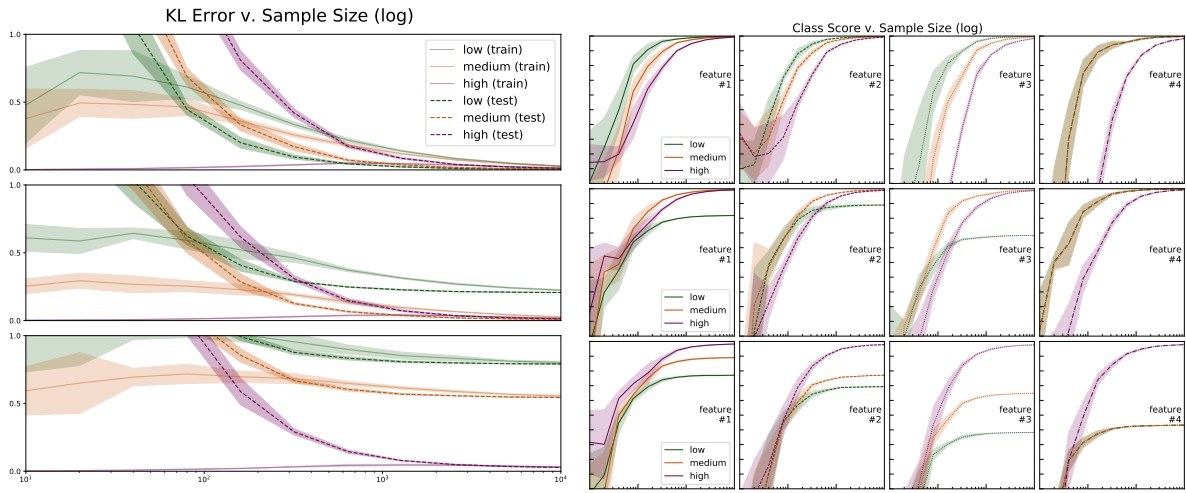

Figure 7: Model Performance vs. Number of Training Samples. Performance evaluated across three different model complexities. Each row correspond to an underlying data distribution which from top to bottom has low complexity, medium complexity, and high complexity. The top row demonstrates the high complexity model overfitting. The bottom row demonstrates the low complexity model underfitting. Left hand side is the KL error objective decreasing; right hand side is the class-wise performance (which is automatically gained from generative performance). Error bars are with respect to 5 different resampling of the synthetic training dataset.

In Figure 7, we show the sample complexity of training when the underlying four-dimensional distribution has low complexity, medium complexity, or high complexity (top to bottom). On the left-hand side, we see the KL error optimized during training, whereas on the right-hand side, we see the calibrated classification score for each of the four dimensions (predicted using the other three features), automatically rising alongside improved generative performance. In the bottom row, we see how the underspecified, low-complexity model leads to underfitting which peaks at subpar performance. In the top row, we see how the overspecified, high-complexity model leads to overfitting which makes less efficient use of the finite dataset (even with multiple thousands of samples). In addition to showing the importance of matching the correct structure

to achieve optimal performance, these experiments also show how achieving good generation performance automatically generalizes to classification performance.

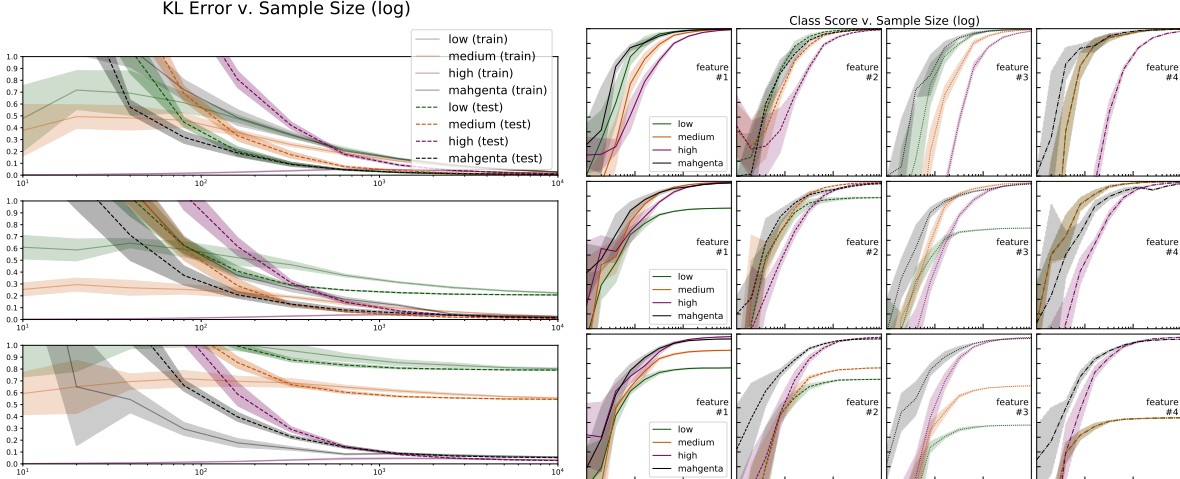

Figure 8: Model Performance vs. Number of Training Samples. Also with Mahgenta automatic selection performance (black) compared against oracle structure performance. As discussed in our theoretical section about overfitting, for low sample sizes, Mahgenta actually outperforms the oracle structure. In most cases, mahgenta very nearly matches oracle performance for larger sample sizes.

### B.3 Real-world Datasets

We next apply our method to the real-world distribution of three different UCI machine learning datasets. Testing split is generated from 50% of the data and kept fixed throughout. The remaining data is split 70%/30% into the training and the validation set. Experiments are run on a Tesla V100 32GB GPU. The three datasets used are shown in Table 1 with their numbers of samples, features, and total possible events.

### B.4 Algorithm Details

.

In Algorithm 2, we give a full description of the subroutines used by our main algorithm.

#### B.4.1 Heredity Styles

Recall that we define the heredity score as:

$$\omega(S) := |\{T : S = T \cup \{i\} \text{ for some } i\} \cap \{T : T \text{ has already been selected}\}| \qquad (23)$$

Thus, writing the count for a particular $S$ as $n_S = \omega(S)$ and its final score as $n_S/|S| = \omega(S)/S$, we can ask what percentage of subsets $T \subseteq S$, of size one less than $S$, are already selected by the algorithm. In the case of pairs, there are the two options of both singles already selected (strong heredity) and only one single already selected (weak heredity). Generalizing to larger $S$ provides much greater flexibility. We focus on a class of heredities defined by a single $\tau\%$ parameter, which asks that the percentage of 'oneless' subsets are at least $\tau$, i.e. $\omega(S)/|S| > \tau$. In the experiments in Section B.1, we consider this for values of $\tau = 30\%, 50\%, 100\%$.

We also consider a heredity style which we call 'semi-strong' which instead asks that $\omega(S) \geq |S| - 1$, meaning that all but one of the 'oneless' subsets have already been selected. For higher-order interactions, more general heredity styles can also be used which consider, for instance, subsets of size two less than $S$. We do not consider these herein.

---

**Algorithm 2:** Full Details of the Mode Interaction Selection Algorithm

---

**1** NEXTAVAILABLEINTERACTIONS(*collection $\mathcal{I}$, heredity strength $\tau$*)

**2**    $\mathcal{J} \leftarrow \emptyset$

**3**    **for** $S \in \mathcal{P}([d])$ **do**

**4**        $n_S \leftarrow \big| \{T \in \mathcal{I} : |T| + 1 = |S|, T \subseteq S\} \big|$

**5**        **if** $n_S/|S| > \tau$ **then**

**6**            $\mathcal{J} \leftarrow \mathcal{J} \cup \{S\}$

**7**    **return** $\mathcal{J}$

**8** TOPINTERACTIONS($\mathcal{J}, K$)

**9**    scores $\leftarrow []$

**10**    **for** $S \in \mathcal{J}$ **do**

**11**        scores$[S] \leftarrow$ SCORE($S$)

**12**    **return** (argsort(scores))$[1 : K]$

**13** GRADIENTDESCENT(*parameters $\Theta$, learning rate $\alpha$, epochs $T$*)

**14**    **for** $t = 1$ **to** $T$ **do**

**15**        **for** $\theta^S \in \Theta$ **do**

**16**            $\theta^S \leftarrow \theta^S - \alpha(-\eta_{\mathrm{trn}}^S + \eta_\theta^S)$

**17**    **return** $\Theta$

**18** MODEINTERACTIONSELECTION($\tau = 30\%, \alpha = 0.50, K = 10, T = 10$)

**19**    $Err_{\mathrm{best}} \leftarrow \infty$

**20**    $\mathcal{I} \leftarrow \{\emptyset\}$

**21**    $\Theta \leftarrow \{0\}$

**22**    **while** $Error(\Theta) < Err_{\mathrm{best}}$ **do**

**23**        $Err_{\mathrm{best}} \leftarrow Error(\Theta)$

**24**        $\mathcal{J} \leftarrow$ NEXTAVAILABLEINTERACTIONS($\mathcal{I}, \tau$)

**25**        $\mathcal{K} \leftarrow$ TOPINTERACTIONS($\mathcal{J}, K$)

**26**        **for** $S \in \mathcal{K}$ **do**

**27**            $\theta^S \leftarrow \vec{0} \in \mathbb{R}^{I_S}$

**28**            $\Theta \leftarrow \Theta \cup \{\theta^S\}$

**29**        $\Theta \leftarrow$ GRADIENTDESCENT($\Theta, \alpha, T$)

**30**    **return** $\Theta$

---

## B.5 Gradient Descent Details

Here we provide the additional necessary implementation details of the gradient descent algorithm. Although the gradient descent algorithm is itself very simple, the bag of practical tricks required to enable scaling to large event spaces quickly becomes very large. Our major innovation is the usage of higher-order Gibbs sampling which allows for much more rapid convergence of MCMC sampling, but we also find the standard usage of annealed importance sampling to provide significant gains. The upsampling of new interaction terms and caching of GPU-computed energy terms provide additional improvements

**Higher-Order Block Sampling** Typical Gibbs sampling constitutes resampling a single coordinate at a time by calculating the conditional distribution of a single variable given all others, $q_\theta(i_k|i_{-k})$. This has been found to be extremely slow for learning higher-order energy models (Min et al., 2014), mainly due to the inability to simultaneously activate all entries of a higher-order energy parameter $\theta^S$. Accordingly, we instead resample according to the conditional distribution of a particular subset of the variables $q_\theta(i_S|i_{-S})$, decidedly choosing $S$ which are already included in our model's collection $\mathcal{I}$. We find that this provides significant speedups over the coordinate-based approach which requires repeatedly sampling from the separate $k \in S$ coordinates before becoming close to the distribution $q_\theta(i_S|i_{-S})$.

**Annealed Importance Sampling** Another critical component of our framework is the usage of annealed importance sampling (Neal, 2001). This allows for the approximation of the normalizing constant $\theta_\emptyset$ without

requiring the sum over billions of elements in the distribution space. This approach also significantly benefits from our development of higher-order Gibbs sampling used during the intermediary steps. We find that this is a critical method for being able to adequately track the progress of the model over time which allows the use of our early stopping procedure during interaction selection.

**Upsampling Active Interactions**  In addition to sampling uniformly across the set of included interactions in the model, we additionally upweight those $\theta^S$ parameters which have only recently been included into the model. This is inspired by the fact that the more recently included parameters are likely to move more quickly during training whereas the older parameters will have already been mostly trained and less mobile. In practice, we use a 50/50 split between the old and new $\theta^S$ parameters, with each round of Gibbs sampling updating the full set of new parameters once and then updating an equal amount of the old, existing parameters.

**Energy Caching**  In conjunction with our technique of sampling multi-dimensional conditional distributions, we find it efficient to avoid redundant computation of the energy functionals. In particular, suppose we are updating the $q_\theta(i_1, i_2 | i_{-12})$, then we will split up our collection as follows: $\mathcal{I} = \mathcal{I}_\emptyset \sqcup \mathcal{I}_1 \sqcup \mathcal{I}_2 \sqcup \mathcal{I}_{12} = \{S \in \mathcal{I} : 1 \notin S, 2 \notin S\} \sqcup \ldots \sqcup \{S \in \mathcal{I} : 1 \in S, 2 \in S\}$. In order to calculate the energies for all possible values of $i_1, i_2$, we only need to calculate the energies once for $S \in \mathcal{I}_\emptyset$ and only for all values of $i_1$ for $S \in \mathcal{I}_1$. We find that caching these values improves the performance while summing over $S \in \mathcal{I}$.

**Practical Warnings**  We close with some practical warnings for reproducibility of the results. It is reminded that the key challenge of doing gradient descent for an energy-based model is the need to calculate the log-partition function $\theta^\emptyset$, but how this translates to the gradient descent algorithm is through poor estimates of the $\eta_\theta$ values, leading to divergence of the training. Fortunately, although it is difficult to know a priori how much Gibbs sampling and importance sampling is required for efficient training, the quick divergence of a training run does easily identify insufficient MCMC sampling (because the convexity of the KL minimization problem ensures the good behavior of gradient descent otherwise). Lastly, it is mentioned that even after these many optimizations, the final MAHGenTa runs on the `mushroom` and `adults` datasets took several days on a single GPU to complete.

### B.6  Verification of Heuristic

Due to the usage of $J(S)$ as a heuristic for measuring the higher-order information in a distribution, we include additional experiments verifying that the MMI and RI values loosely track one another on most distributions. In Figure 9, we sample 100 random distributions for dimensions 3, 4, and 5, computing the refined information of the corresponding dimension, alongside the multiple mutual information for that distribution. Finally, a scatter plot of the 100 distributions is provided in Figure 9 and the Pearson correlation is calculated to be: 0.8741, 0.7563, and 0.6376.

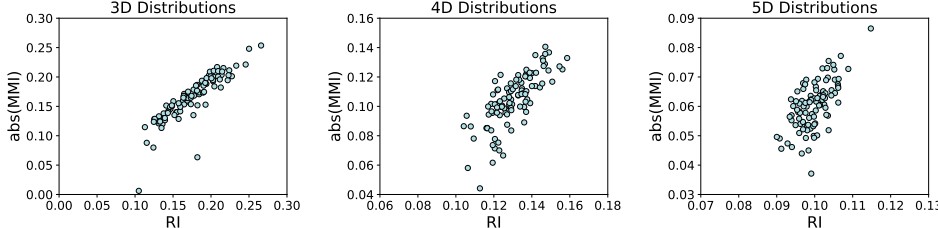

Figure 9: For 100 random distributions of sizes $d = 3, 4, 5$ (with $I = 5$), we plot the absolute value of the multiple mutual information against the marginal refined information, finding Pearson correlation coefficients of 0.8741, 0.7563, and 0.6376.

## C   Further Theoretical Discussion

### C.1   Order Invariant Values

Since there is an abundance of complete, supported chains on the lattice of $\mathcal{P}([d])$, we may still be interested in some canonical information of a mode-interaction, without too much consideration of its supporting information set. Accordingly, let us define two canonical values of the information of a set $S$ not associated with a specific chain or interaction set. That will be the 'marginal refined information', given by using the minimal supporting information set and the 'conditional purified information', given by using the maximal supporting information set.

$$RI_S^{\mathrm{marg}} := RI_{\mathcal{I}_S^{\min},S} \qquad\qquad \mathcal{I}_S^{\min} := \mathcal{P}(S) - S$$
$$RI_S^{\mathrm{cond}} := RI_{\mathcal{I}_S^{\max},S} \qquad\qquad \mathcal{I}_S^{\max} := \mathcal{P}([d]) - \{T : T \supseteq S\}$$

We note that the mutual information corresponds to the marginal refined information in the case that $|S| = 2$. In other words, $\mathrm{MI}(X_i, X_j) =: I(\{i,j\}) = RI_{\{i,j\}}^{\mathrm{marg}} = RI_{\{\emptyset,i,j\},\{ij\}} = RI_{\{\emptyset,i,j\}\to\{\emptyset,i,j,ij\}}$.

### C.2   Relation to Causal Structure Learning

We further make clear the relationship to structure learning with a simple example in causal structure learning. Suppose we have the distribution as induced by the causal graph depicted in Figure 10. Although there is mutual information between the variables $B$ and $C$, there is no direct causal information between them. In fact, they are both controlled by the variable $A$ and the correlations between them are thereafter induced.

In the case of $d = 3$ and $|S| = 2$, the refined information only takes two values (which correspond to the marginal and conditional values). In Figure 10, we have these values calculated for all three pairs of variables. If we take a look at the mutual information between $B$ and $C$, we can indeed see there is a positive amount of information; however, in the presence of conditioning on $A$, there is no refined information between these two variables.

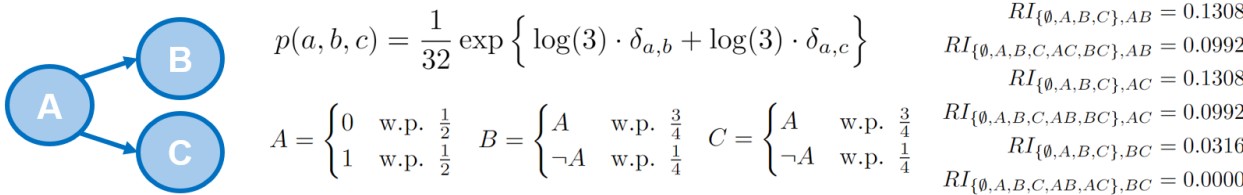

$$p(a,b,c) = \frac{1}{32} \exp\left\{ \log(3)\cdot\delta_{a,b} + \log(3)\cdot\delta_{a,c} \right\}$$

$$A = \begin{cases} 0 & \text{w.p. } \frac{1}{2} \\ 1 & \text{w.p. } \frac{1}{2} \end{cases} \quad B = \begin{cases} A & \text{w.p. } \frac{3}{4} \\ \neg A & \text{w.p. } \frac{1}{4} \end{cases} \quad C = \begin{cases} A & \text{w.p. } \frac{3}{4} \\ \neg A & \text{w.p. } \frac{1}{4} \end{cases}$$

$$RI_{\{\emptyset,A,B,C\},AB} = 0.1308$$
$$RI_{\{\emptyset,A,B,C,AC,BC\},AB} = 0.0992$$
$$RI_{\{\emptyset,A,B,C\},AC} = 0.1308$$
$$RI_{\{\emptyset,A,B,C,AB,BC\},AC} = 0.0992$$
$$RI_{\{\emptyset,A,B,C\},BC} = 0.0316$$
$$RI_{\{\emptyset,A,B,C,AB,AC\},BC} = 0.0000$$

Figure 10: Simple causal graph to help illustrate refined information.

Although the traditional tools of causal structure learning, namely a set of conditional independence tests, are sufficient to identify the causal structure in this case (or at least the Markov equivalence class), the tool of refined information is imagined as an even more powerful tool which may distinguish additional higher order interactions between variables and various hypergraphical extensions to existing graphical models.

### C.3 Structure of Mode Interactions

Here we provide some additional figures to more quickly provide intuition about the algebraic structures we introduce. In Figure 11, we depict a possible chain which is maximally refined for $d = 3$.

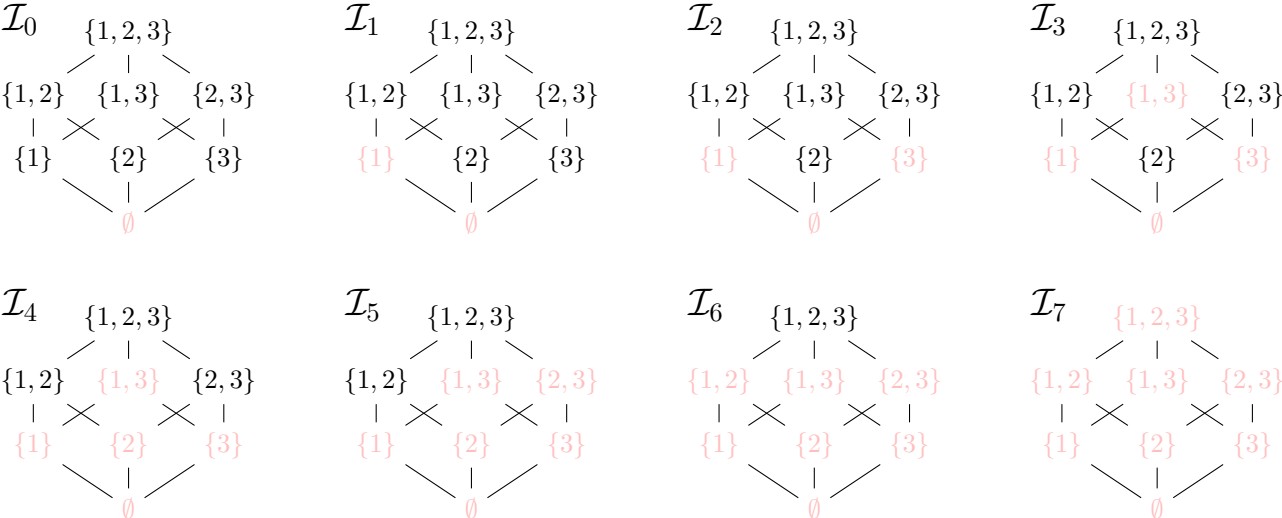

Figure 11: Example of a possible chain of mode interaction collections.

In Figure 12, we depict the algebraic structure representing all possible hierarchical chains which could be selected. Each mode interaction $S \subseteq [d]$ is represented by a different color and all arrows are drawn which are possible to add while still obeying the hierarchical condition. Horizontally sorted by the size of each collection $\mathcal{I}$, although redundancies are suppressed (e.g. $\{\emptyset, \{1\}, \{2\}, \{1, 2\}\} = \{\emptyset, 1, 2, 12\}$ is written as $\{12\}$).

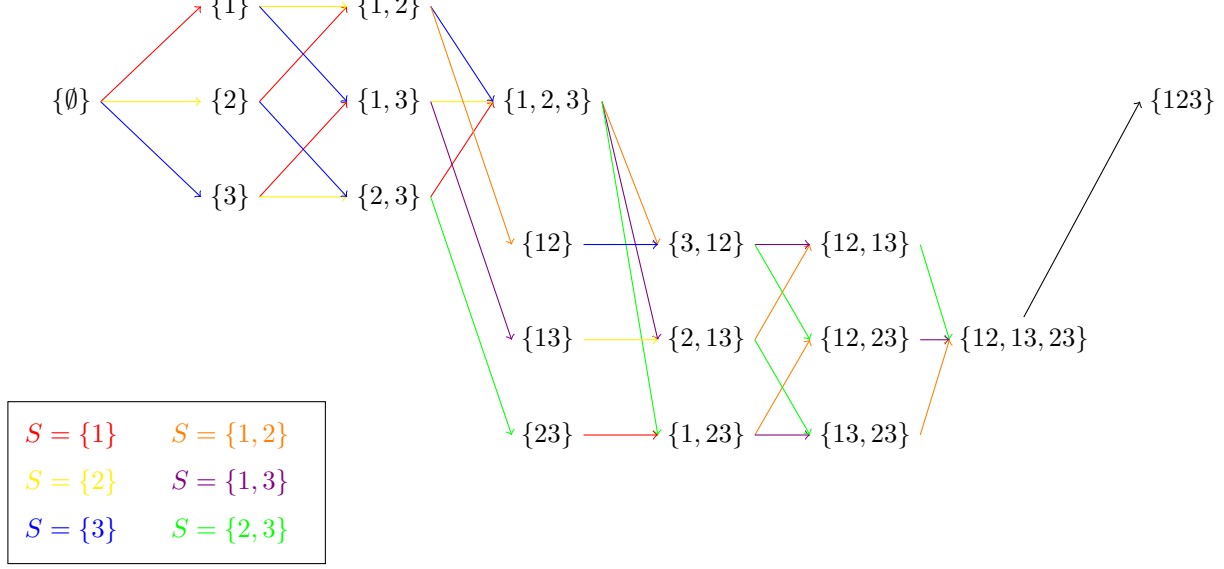

Figure 12: Algebraic structure of all hierarchical collections of mode interactions.

## C.4 Learning Problem Formulation

In its simplest form, distribution learning is about modeling a distribution $q$ which accurately matches the true distribution $p^*$. In this work, we use the objective of forward KL divergence which is equivalent to the maximum likelihood approach.

$$D_{KL}(p^*; q) = \sum_i p^*(i) \cdot \log\left(\frac{p^*(i)}{q(i)}\right) = \sum_i p^*(i) \cdot \log\left(p^*(i)\right) - \sum_i p^*(i) \cdot \log\left(q(i)\right),$$

$$\min_q \left\{ D_{KL}(p^*; q) \right\} = \max_q \left\{ \sum_i p^*(i) \cdot \log\left(q(i)\right) \right\}.$$

Importantly, as discussed throughout the paper, we reframe this objective via the choice of a sparse set of mode interactions in order to achieve better generalization properties from the learned distribution. In particular, we may write that:

$$D_{KL}(p; \hat{q}_{\mathcal{I}}) = D_{KL}(p; p_{\mathcal{I}}) + D_{KL}(p_{\mathcal{I}}; \hat{q}_{\mathcal{I}}).$$

This means that, after a choice of interactions $\mathcal{I}$, our KL error decomposes orthogonally into an estimable part and an inestimable part. This means that a choice of small $\mathcal{I}$ will have a large inestimable error but will very accurately predict the estimable part of the distribution, whereas a choice of large $\mathcal{I}$ will have a small inestimable residual but will have a much more challenging distribution to estimate directly.

Accordingly, we may write our complete objective as the following bilevel optimization problem with an outer combinatorial search over the space of $\mathcal{I}$ and an inner continuous optimization over the $\theta_{\mathcal{I}} = \{\theta^S\}_{S \in \mathcal{I}}$ parameters:

$$\min_{\mathcal{I}} \left\{ \min_{\theta_{\mathcal{I}}} \left\{ D_{KL}(p; \hat{q}_{\theta_{\mathcal{I}}}) \right\} \right\} = \min_{\mathcal{I}} \left\{ D_{KL}(p; p_{\mathcal{I}}) + \min_{\theta_{\mathcal{I}}} \left\{ D_{KL}(p_{\mathcal{I}}; \hat{q}_{\theta_{\mathcal{I}}}) \right\} \right\}.$$

As discussed, the inner optimization is handled via a gradient descent algorithm which leverages multiple necessary Monte Carlo approaches in order to learn the distribution parameters while handling the intractability of the normalizing constant. The outer combinatorial optimization is handled with a simple greedy heuristic. The major benefit of using a greedy heuristic is the fact that our search across the space of $\mathcal{I}$ will proceed along a single hierarchical chain. It follows that the inestimable portion will be monotonically decreasing along our chain and the remaining error from the estimable part will only increase as we continue to increase the complexity of the learned distribution with our fixed and finite number of samples. This allows us to fit snugly within the 'classical' regime of overfitting where we have at our disposal the simple rule of stopping as soon as our validation error no longer improves.

This results in our final approach of using

$$\operatorname*{argmin}_{\mathcal{I}, \theta_{\mathcal{I}}} \left( D_{KL}(p_{val}; \hat{q}_\theta^{\mathcal{I}}) \quad \text{where} \quad \hat{q}_\theta^{\mathcal{I}} = \operatorname*{argmin}_{q_\theta \in \mathcal{M}_{\mathcal{I}}} \left( D_{KL}(p_{trn}; q_\theta^{\mathcal{I}}) \right) \right)$$

to validate the learned distribution on a subset of data which is different from the training samples which are used to fit the continuous $\theta$ parameters.

