# OpenReview forum: "A Complete Decomposition of KL Error using Refined Information and Mode Interaction Selection"
_TMLR — Accepted by TMLR_

### Review · Reviewer_Y1SM · 2025-12-25

**Summary Of Contributions:**

This paper revisits hierarchical log-linear (energy-based) models for discrete/tabular distribution learning through an information-geometric lens. Specifically, the authors focus on higher-order interactions beyond standard pairwise Boltzmann machines and Markov graphical models. The main contributions of this work include:

* Refined Information (RI): Introduces a notion of "refined information" for interaction sets, which is defined via KL differences between information-geometric projections onto nested hierarchical submanifolds. This yields a complete additive decomposition of $D\_{KL}(p;u)$ (relative to uniform) along a chosen maximally refined hierarchical chain.

* Mode Interaction Selection: Uses the KL-decomposition viewpoint to motivate a sparse interaction selection problem for hierarchical log-linear models, with an early-stopping argument to control overfitting along a monotone capacity growth process.

* MAHGenTa algorithm: Proposes an end-to-end practical system combining (1) a greedy Mode Interaction Selection heuristic that uses $|J(S)|$ as a computable proxy for RI gains, and (2) GPU-based training of hierarchical log-linear models with purified gradients plus higher-order block Gibbs sampling and annealed importance sampling to handle normalization and likelihood tracking.

* Empirical evaluation: Demonstrates improved KL/log-likelihood vs 1-body and 2-body baselines on synthetic data and several UCI datasets, and shows that good generative modeling yields useful downstream conditional prediction/classification performance.


Strengths:

* Clear motivation for going beyond pairwise structure; the XOR example highlights why pairwise MI can miss genuine high-order dependence.
* The projection/KL-drop definition makes the “information attribution” nonnegative by construction, which is conceptually appealing.
* The paper attempts to bridge theory and practice for higher-order energy-based models. Empirical results indicate consistent gains in generative fit from adding selected higher-order interactions.


Weaknesses/limitations:

* The algorithm uses $|J(S)|$ as a proxy for refined information, but the paper appears to offer limited direct evidence that this proxy reliably tracks the true RI gain in realistic settings, while the success of Mode Interaction Selection may depend critically on this correlation.
* Comparisons are mainly against 1-body/2-body log-linear models; it is unclear how MAHGenTa compares to other scalable discrete generative baselines (e.g., autoregressive models for tabular data, modern neural density estimators, or other higher-order sparse energy-based methods).

**Audience:**

Yes

**Audience Explanation:**

This paper studies the theoretical frameworks and practical methods for learning and understanding distributions, especially for discrete data and energy-based models, which are relevant to the TMLR community.

**Claims And Evidence:**

Yes

**Claims Explanation:**

Theoretical definitions (refined information via KL differences between successive projections) are internally consistent. The conceptual claims about KL decomposition and attribution are supported. Empirical results on synthetic data and several real-world datasets show the improvement made by MAHGenTa with KL (or log-likelihood proxy) over independent and pairwise Boltzmann baselines, supporting the claim that selected higher-order interactions can improve generative performance.

**Requested Changes:**

* Validate the MIS scoring proxy vs true refined information: It would benefit if the authors could provide experiments showing that $|J(S)|$ correlates with the actual RI gain or with true KL improvement.

* Comparisons against other scalable discrete generative baselines, e.g., autoregressive models for tabular data, modern neural density estimators, or other higher-order sparse energy-based methods.

---

> ### Author Response · Authors · 2026-02-11
>
> We first want to thank you for your review of our work.  We believe you have identified the key contributions of our work as “Clear motivation for going beyond pairwise structure” which is solved by refined information; “the projection/KL-drop definition makes the “information attribution” nonnegative by construction, which is conceptually appealing”; and the MAHGenTA algorithm as an end-to-end practical system which “attempts to bridge theory and practice for higher-order energy-based models”.
>
> Alongside these many strengths, the weaknesses which you are focused on addressing are (1) the usage of $|J(S)|$ as an appropriate heuristic and (2) the comprehensiveness of the evaluation, especially with respect to modern neural density estimators or other higher-order sparse energy-based methods.  We address each of these concerns individually.
>
> > The algorithm uses as a proxy for refined information, but the paper appears to offer limited direct evidence that this proxy reliably tracks the true RI gain in realistic settings, while the success of Mode Interaction Selection may depend critically on this correlation.
>
> We begin by first trying to qualify this claim to ensure this is not a miscommunication due to bad wording on our part.  We want to start by saying that we do not want to explicitly claim that MMI tracks the RI value.  Every existing interaction selection algorithm requires two ingredients: a heredity measurement to allow exploration (without exponential cost) and a heuristic to guide this exploration.  Thus, when we want to use MMI as this heuristic, we are not claiming it is tracking RI, but rather that the search ordering provided by MMI is sufficiently similar to the ordering of RI.  We have updated the wording to make sure this is more clearly communicated.
>
> We note that this is not only a weaker condition, but also that there are very few alternatives.  Seemingly, the only alternative would be the question of whether or not a random ordering could outperform this heuristic.  We indeed did not compare with this baseline, please let us know if this is what you have in mind.  In order to nonetheless show that the MMI has some relationship with RI, we have run an experiment which calculates the MMI and RI for small random distributions and compares them, finding a Pearson correlation of 0.87, 0.76, and 0.64 on 3D, 4D, and 5D datasets.
>
> > Comparisons are mainly against 1-body/2-body log-linear models; it is unclear how MAHGenTa compares to other scalable discrete generative baselines (e.g., autoregressive models for tabular data, modern neural density estimators, or other higher-order sparse energy-based methods).
>
> Unfortunately, the difficulty of directly embedding this work’s contributions to information theory and energy-based modeling within the modern directions of machine learning seems to be an unavoidable limitation.  In particular, works on generative machine learning and variational inference have long since discarded achieving exactly likelihood values (equivalently exact KL values), making a direct comparison on KL metrics impossible.  We attempted to emphasize this point in the “SOTA Generative Models” section of the related work.
>
> Given that we have already reviewed existing discrete and likelihood-based approaches, is it possible to be more specific on which neural-based density estimators or autoregressive tabular approaches you believe are feasible for comparison here and what comparison experiments you think would be necessary for this?  We are happy to investigate this further; however, it is our current understanding that it is not possible to directly compare.  It is our understanding that we significantly advance the state-of-the-art when restricting to these likelihood-based approaches.  Beyond this, we also hope this work can position itself as providing the fundamental theoretical tools for understanding the generalizability of any generative model.  These types of tools do not seem to exist anywhere throughout current machine learning theory.

---

### Review · Reviewer_eaNw · 2026-01-06

**Summary Of Contributions:**

When learning probability distributions over discrete variables, log-linear models such as Boltzmann machines and Markov random fields are widely used. However, most of these models are limited to pairwise (two-body) interactions. Incorporating higher-order interactions (among three or more variables) has been avoided due to increased model complexity and the difficulty of structure learning.
This paper employs the framework of information geometry to quantify how much purely new information a given set of interactions contains, i.e., information that cannot be explained by lower-order interactions. The authors claim that the decrease in KL divergence (error) resulting from adding a new interaction to the model can be completely and non-redundantly decomposed into a sum of refined information contributed by each interaction.
Based on this principle, they propose a sparse structure learning algorithm, which sequentially selects interactions that most efficiently reduce the KL error.
Because it is based on a theoretically guaranteed error decomposition, it provides a clear criterion for which interactions to add. Consequently, the method can efficiently represent complex data structures that conventional second-order models (e.g., Boltzmann machines) fail to capture. Experiments on both synthetic and real datasets demonstrate that the proposed method achieves higher performance with fewer samples compared to existing approaches.

**Additional Comments:**

Regarding the non–self-consistent explanation of information geometry, the following improvements could be made. These would help increase the transparency of the theory and assist readers in developing an intuitive understanding of why the KL error can be decomposed so cleanly.

While I do not claim that these revisions are strictly mandatory, the authors should be aware that the current presentation is not sufficiently convincing in its present form.

* At the beginning of Section 3.2 or 3.3 (not in the Appendix in its current form), the Pythagorean theorem for the KL divergence should be introduced explicitly, together with an illustrative figure. In doing so, the notion of orthogonality should be clearly defined: namely, that e-flat manifolds (exponential families) and m-flat manifolds (mixture families) are orthogonal, and that this orthogonality is precisely the condition under which the Pythagorean theorem holds. This is partially discussed in Appendix C.5, but a visual and intuitive explanation in the main text would be highly desirable.

* In the context of the hierarchical model proposed in this paper, it should be further explained why each projection step satisfies this orthogonality condition, based on the inclusion relationship between manifolds (i.e., $M_{I_{t-1}} \subset M_{I_{t}}$).

* The “chain of collections” introduced in Section 3.3 could be visualized not merely as a sequence of sets, but as a path in the space of probability distributions. This would allow the procedure of e-/m-projections to be explained more concretely. Beyond the algebraic structure shown in Figure 10 and 11, one could depict how the distribution moves geometrically in probability space—from the uniform distribution to the target distribution—as specific interactions are added step by step, represented as a sequence of points and projections on manifolds. In this visualization, it would also be helpful to explain that different choices of ordering (i.e., different paths) can lead to different values of refined information at each step, interpreted geometrically as differences in the “route” taken through the space.

* Finally, regarding the purified gradient introduced in Section 4.2, it would strengthen the presentation to explain it not merely as a computational trick (invariance under constant shifts), but as an operation that removes redundancy in the parameter space and corresponds to a well-defined operation on the statistical manifold. This would further improve the consistency between the optimization procedure and the information-geometric framework developed in the paper.

**Audience:**

Yes

**Audience Explanation:**

Inference methods for energy-based models with higher-order interactions are important, and the proposed approach is likely to be of interest to a substantial portion of the TMLR readership. In particular, the method clearly quantifies the contribution of each interaction to the error (in terms of KL divergence), and it also discusses potential applications of this perspective, which strengthens its appeal.
Especially, the presentation in Figure 3 is excellent. In addition, the application to fairness discussed at the end of Section 5.3 is highly interesting.

**Broader Impact Concerns:**

N.A.

**Claims And Evidence:**

No

**Claims Explanation:**

First, regarding the information geometry that forms the core of the proposed method, although the procedures themselves are described in a self-contained manner, it is difficult to conclude that the paper sufficiently explains why this approach works.

In particular, to gain geometric intuition for why the KL divergence can be decomposed into a simple sum, analogous to the Pythagorean theorem, one needs external knowledge about the dual flat structure and foliation of statistical manifolds. Without such background, the exposition feels like a conceptual leap. Especially, the idea that $\mathcal{D}(p | \Pi_{\mathcal{I}_k}(p))$ can be decomposed as a sequence of orthogonal projections—the geometric concept underlying Figure 1 and Figure 2—is likely to be very difficult for first-time readers to grasp. In this sense, the current presentation seems accessible only to a very limited audience, and I cannot judge that the claims are supported by "convincing and clear evidence".
I will describe possible improvements to address this issue later.

In addition, there are several other points where the exposition is unclear. These will also be discussed below.

**Requested Changes:**

* In the paper, for computational convenience, $R_I$ (refined information) is heuristically replaced by $J_S$ (a measure of higher-order interactions). However, an explanation bridging this gap is necessary. In particular, the paper lacks a justification of this approximation: it should be explained how $J_S$ relates to local coordinate changes in information geometry, or in what sense $J_S$ provides a good approximation to $R_I$ under small perturbations. A detailed explanation of the rationale and validity of this approximation is required.

* In Section 4.2, the phrase “We first reiterate the gradient of a log-linear model...” is used. Since this is the first appearance of the gradient in the paper, “reiterate” is inappropriate and should be revised.

* Please provide a derivation of Equation (7).

* Regarding the clarity of the paper, the description of “our hyperparameters” in Section 5.2 is insufficient. Even including the appendix, it is unclear which specific hyperparameters were swept (e.g., learning rates, regularization terms, etc.). These details should be stated explicitly.

---

Note: If addressing the requested changes leads to page limit issues, it may not be necessary to devote a full paragraph each to Higher-Order Block Sampling, Upsampling Active Interactions, and Annealed IS in Section 4.2. It would be sufficient to briefly state that these techniques are used, with detailed explanations deferred to the appendix.

---

> ### Author Response · Authors · 2026-02-11
>
> Thank you for your review.  We apologize that unclear exposition for providing the necessary background made you feel that our work did not sufficiently support the claims we made, despite its appeal to a substantial portion of the TMLR readership.  It sounds like you are mainly frustrated with the conceptual leap required to embrace the information geometry perspective and the lack of self-contained reproductions of the necessary theorems.  We thank you for pushing the writing forward to ensure it is accessible to the widest audience possible.  We address these concerns below.  That being said, based on your questions, it seems like you were able to understand the key concepts from information geometry which we applied in this work.
>
> We will discuss the improvements we made to the information geometry presentation as well as our answers to your questions on our algorithm and experiments, but first we do feel the need to be picky about a single line in your summary to ensure there is no miscommunication: “Because it is based on a theoretically guaranteed error decomposition, it provides a clear criterion for which interactions to add.”  Here, we want to make sure it is understood that our theoretical contribution of refined information and the KL error decomposition, must be detached from our practical contributions, which are the MIS selection and MAHGenTa algorithm.  In particular, although refined information exactly describes the drop in KL error (for a chosen ordering), it is not easy to compute, and so therefore *it would provide* a clear criterion, however, in practice *it does not provide* and we must use some alternative choice for choosing the ordering.  We discuss this further below.
>
>
> ## Presentation of Information Geometry
>
> We have completely reworked the Information Geometry section in the appendix to hopefully make it an easier read for TMLR audience members without any previous exposure to these concepts.  We hope that this addresses the concerns you labeled and please let us know if there are any concerns remaining.  We have also made some slight changes in Section 3 to make the reading smoother overall.
>
> Based on your questions, we feel you have already demonstrated a good overall understanding of how we use information geometry to create a chain of distributions.  We address some of the particular questions which may require a more nuanced explanation.
>
> > The “chain of collections” introduced in Section 3.3 could be visualized not merely as a sequence of sets, but as a path in the space of probability distributions
>
> Yes, you are exactly right that after fixing a distribution $p$, the chain of collections directly corresponds to a sequence of probability distributions, but moreover a sequence of distributions which form a path connected by geodesics which are orthogonal to each other at the points along the path.  Slightly differently, when considering the entire manifold (all possible $p$), the chain of collections would then corresponds to a particular fibration of the entire manifold, i.e. all possible paths at once.  We have also added a sentence to the beginning of 3.3 to make this intuition apparent immediately.
>
> > In the context of the hierarchical model proposed in this paper, it should be further explained why each projection step satisfies this orthogonality condition
>
> Intuitively, each projection step is orthogonal because considering the three distributions P, Q, R: Q will match the $\eta$ coefficients of P on a subset and match the $\theta$ coefficients of R on the complement, meaning that the orthogonality condition holds (see Pythagorean and projection theorems now provided in the appendix.)

---

> > ### Author Response · Authors · 2026-02-11
> >
> > > In this visualization, it would also be helpful to explain that different choices of ordering (i.e., different paths) can lead to different values of refined information at each step, interpreted geometrically as differences in the “route” taken through the space.
> >
> > Although we agree that a visualization for this would be very nice, there are some practical challenges in combining the manifold visualization of paths with the algebraic visualization of paths in Figure 12.  In particular, for $d=2$, although binary variables have 3 total dimensions making it visualizable, there are only two possible paths.  For $d=3$, there are many more paths like in Figure 12, but even binary variables already have 7 total dimensions, making it very hard to provide a visualization which is not geometrically misleading.
> >
> > > Please provide a derivation of Equation (7).
> >
> > We have already provided a derivation of Equation 7 (now Equation 8) in the Appendix, now in App A.3.  This equation is the gradient which corresponds to the centered coordinates we introduce as an alternative to remove the redundancy of the parameters.  Importantly, we are attempting to simultaneously emphasize the removal of the redundancy alongside the better computational properties of the centered coordinates (which is distinct from previous information geometry approaches which do not focus on this aspect.)  If there is some wording which is making this unclear, please let us know.
> >
> >
> > ## Interaction Selection Algorithm
> >
> > Regarding the interaction selection algorithm, we hope that our above comments on your summary “Because it is based on a theoretically guaranteed error decomposition, it provides a clear criterion for which interactions to add” help to clarify the need to separate our theoretical and practical contributions.  In particular, although the refined information decomposition provides a theoretical justification for using the early stopping procedure which is empirically validated in Figure 4, it unfortunately does not provide direct guidance on which interactions to add.  More specifically, the only current algorithm for calculating refined information is the gradient descent approximation which we introduce herein.  Because of this, we cannot use the result of the gradient descent algorithm for the choice of which interaction to be fit by the gradient descent algorithm.  We hope this begins to help clarify questions related to the algorithm.
> >
> > > Regarding the clarity of the paper, the description of “our hyperparameters” in Section 5.2 is insufficient.
> >
> > We have added the list of hyperparameters to the main body of the paper.  In those experiments, we only change the MIS hyperparameters, like the heredity strength and the heuristic renormalization, as well as the loss function of true likelihood or pseudo-likelihood.  We do not vary the gradient descent hyperparameters like learning rate.  This comparison to pseudo-likelihood is used to enable a fairer comparison to the two previous works [Schmidt & Murphy,2010] and [Min et al., 2014] which do not have publicly available code (and were only implemented for binary variables, anyways).  Both those works use pseudo-likelihood, but we show using the true likelihood is superior by a significant amount.
> >
> > > In the paper, for computational convenience,  (refined information) is heuristically replaced by  (a measure of higher-order interactions). However, an explanation bridging this gap is necessary.
> >
> > We hope this heuristic choice is now better understood given our explanation that, in general, refined information can only be calculated through the use of a gradient descent algorithm.  In that sense, we do not want to specifically claim that $J_S$ approximates $RI$ under small perturbations or otherwise.  We only claim, implicitly, that the ordering provided by $J_S$ is more informative than a completely random order.  We hope this much weaker claim is more believable than the claim that $J_S$ approximates $RI$.  Nevertheless, we have provided additional experiments in App B.6 showing that $J_S$ generally correlates with $RI$.
> >
> > If there are any concerns regarding our claims or the presentation of our claims which remain after these edits and our response, please let us know.

---

### Review · Reviewer_ZRNL · 2026-01-29

**Summary Of Contributions:**

Summary of Contributions

This manuscript introduces a novel information-geometric framework to decompose the KL error of discrete probability distributions into a sum of non-negative contrbutions termed 'refined information'. This theoretical formulation enables a procedure for progressive selection of interaction sets for hierarchical log-linear models, allowing them to capture higher-order variable interactions that are out of the scope for (fully-visible) Boltzmann machines or Markov models. The authors develop MAHGenTa, a hierarchical log-linear algorithm that utilizes a greedy heuristic based on Multiple Mutual Information to capture higher-order interactions between discrete variables. The work demonstrates the feasibility of this framework on some synthetic datasets and simple tabular real-world dataset, showing that this higher-order generative models can be used for multi-target classification tasks and perform comparably to classical binary classification baselines such as logistic regression.

Strengths:
- provides a mathematically rigorous framework that projects the data distributions into flat manifolds, ensuring that the optimization problem of minimizing the KL divergence is convex.
- "refined Information" is a positive and additive metric that provides a novel decomposition of KL divergence, and allows for a more interpretable selection of interaction subsets.
- proposes an efficient sampling strategy as an alternative to Gibbs sampling
- the GPU implementation allows to scale the approach to to real-world tabular data, and evaluations show that the generative training procedure allows a single model to be used for multi-target classification without retraining.

Weaknesses
- the presentation of visual data is lacking, with main text figures missing axis labels and descriptive captions.
- heavy reliance on the Appendix and reference for fundamental theoretical and implementation details.
- several sections suffer from unclear phrasing and grammatical inconsistencies that hinder the readability of the theoretical arguments.

Overall the paper is certainly interesting, but given the density of information and the page limit of TMLR, I am not quite sure if this is the right venue for this work.

**Audience:**

Yes

**Audience Explanation:**

Yes, the paper is likely to interest researchers working on problems related to information geometry, statistical mechanics, and tabular data modeling. For example, the bridge between many-body expansions in physics and hierarchical log-linear models in machine learning provides a compelling perspective for those working on interpretable generative models or the modelling physical systems using machine learning. Additionally, the demonstration of using generative energy-based models for simultaneous classification tasks and bias inspection in sensitive features (e.g., race/gender) aligns with current interests in explainable AI and algorithmic fairness.

**Broader Impact Concerns:**

No concerns.

**Claims And Evidence:**

Yes

**Claims Explanation:**

The theoretical claims claims regarding the non-negativity and additivity of Refined Information, as well as the convex nature of the optimization procedure are well-supported by the mathematical definitions provided.
The experimental evidence on synthetic datasets effectively demonstrates the expected underfitting-overfitting behavior associated with different model complexities.

However, the criterium of clear evidence is somewhat undermined by the lacking presentation of the theoretical framework, as well as the poor presentation of some figures, particularly figures 2 and 3, which lack any labeling of the axes, which paired with the uninformative captions, which is not self-sufficient at all, make it unnecessarily difficult to understand and interpret the figures.
Furthermore, the comparison to existing sparse graphical models makes sense within the context of this paper, but makes it difficult to place the relevance of this work within the context of contemporary machine learning research. Maybe some benchmarks about computation time can give a sense of how feasible is it to apply aspects of this work to more large scale datasets.

**Requested Changes:**

- The lack of clear labeling and captioning of the figures is a big detriment to the paper. Add clear labels to the x and y axes for Figures 2, 3, and 4. Ensure that captions provide a self-contained explanation of what is being shown in the figure (critical).

- Some more detailed and rigorous evaluations are needed to contextualize the utility of the model compared to other approaches, as well as empirically demonstrate the actual utility of some of the design choices of the framework. Provide a computation time benchmarking of the model for the different datasets, and compare it to the competing approaches. Also include some ablations with regards to the more critical components such as a comparison of the benefit (whether for accuracy or computation time) of the block sampling procedure, or a comparison on some toy examples of the differences in selection of interaction collections between the theoretical RI and mutual information (critical).

- clearly refer to the the definition and explanation of "strong" and "weak" heredity from the Appendix into Section 4.1 to ensure the selection algorithm is reproducible from the main text, or ideally explain it in plain words in the main text since it's currently lacking a textual explanation.

- correct grammatical errors and awkward phrasing throughout the manuscript. Some examples:
* Page 2, Section 2.1: "modern feature selection approaches mainly concern themselves with paying the proper credence ", paying credence does not work, it is usually phrased as giving credence.
* Page 3, Section 2.3: "The most related work to ours is the few works that have extended the sparse graphical modeling formulation...", awkward because of mixup of plural and singular
* Page 10, Section 5.3: "however, in our energy model working directly on the observed variables, it is made explicit the learned connections between variables.", an 'in' seems to be missing.

---

> ### Author Response · Authors · 2026-02-11
>
> Thank you for your review, we appreciate your identification of the major strengths of our paper:
>
> - a mathematically rigorous framework, ensuring that minimizing the KL divergence is convex.
> - refined information and a novel decomposition of KL divergence
> - an efficient sampling strategy as an alternative to Gibbs sampling
> - the GPU implementation allows to scale the approach to to real-world tabular data
> - experimental evidence on synthetic datasets effectively demonstrates the expected underfitting-overfitting behavior
>
> Below, we first address your concerns on the clarity of presentation and then address your concerns on experiment details and overall positioning.
>
> ## Presentation Concerns
>
> We have made changes throughout the manuscript to improve the wording and clarity of presentation.  Beyond the specific sentences you indicated, we have tried to fix unclear wording in many places and also completed some reorganization of the presentation of the information geometry theory which we use.  Hopefully these make the paper overall much easier to understand.  If you feel there are concerns you had which we did not address in this updated version, please let us know.
>
> > The lack of clear labeling and captioning of the figures is a big detriment to the paper.
>
> We have updated the captions for Figures 2 and 3, so hopefully the data presented is now much easier to understand.  If there are still any figures throughout which remain unclear, please let us know so that we can explain the figure and also make the necessary changes to improve the presentation of the data.
>
> > clearly refer to the the definition and explanation of "strong" and "weak" heredity from the Appendix into Section 4.1
>
> We have now added a definition of heredity strength directly in Section 4.1, but also a further discussion in Appendix B.4.  The original strong and weak heredity only applies to pairs, and says that we should consider the subset {i,j} only if both {i} **and** {j} are already included in the model (strong heredity) or only if {i} **or** {j} are already included in the model (weak heredity).  This method of exploring the interaction sets is necessary to avoid the exponential explosion of considering all possible $S$.  This is then combined with the heuristic in order to do an efficient search.
>
>
>
> ## Experiment Concerns
>
> Overall, we get the impression that you feel that although our theoretical contribution with refined information and mode interaction selection is solid, bar potential issues in presentation, you feel that the practical contribution of the MAHGenTa algorithm is not sufficiently verified by the experimental results we show.  In particular, although we achieve SOTA performance amongst likelihood-based approaches using energy-based modeling, going beyond what was possible in the two most closely related works, you feel these contributions are disconnected from current machine learning practice in generative modeling, making you feel that insufficient experiments have been performed.  We will try to address your individual points to explain why we feel the experiments performed already provide empirical justification for the theoretical contributions we made; however, we are open to further discussion on what experiments could be necessary for making the MAHGenTa algorithm relevant for a wider audience.
>
> > the comparison to existing sparse graphical models makes sense within the context of this paper, but makes it difficult to place the relevance of this work within the context of contemporary machine learning research
>
> Unfortunately, we feel that this rift between contemporary machine learning practice and theoretical understanding of generative models may be difficult to rectify.  It is our impression that specifically with respect to likelihood-based probabilistic modeling approaches, there are no alternatives which can scale to the dataset sizes which we apply MAHGenTa to.

---

> > ### Author Response · Authors · 2026-02-11
> >
> > This gap is what we attempted to address in the “SOTA Generative Models” section from the related work.  To be explicit, modern models like VAEs, GANs, and diffusion never compute a closed-form likelihood function, making it impossible to directly compare on a metric like KL divergence.  Although those literatures have come upon with an abundance of new computable metrics to compensate for this issue, they are commonly only applicable to a specific domain, e.g. Fréchet Inception Distance (FID) and Inception Score (IS).  In terms of positioning, we instead hope that our work can be considered as a theory contribution which is paired alongside a theoretically-motivated algorithm which nonetheless advances state-of-the-art for energy-based modeling.
> >
> > > Maybe some benchmarks about computation time can give a sense of how feasible is it to apply aspects of this work to more large scale datasets.
> >
> > We have added some additional discussion to the appendix regarding the practical considerations of using the gradient descent training algorithm.  For the two larger datasets, our final model was trained on a single GPU over several days.  Continuing to scale up these methods is a constant and unavoidable challenge, which is why we intend to release our code alongside the paper for anyone needing to compute refined information.  Moreover, it can be imagined that in the same way that mutual-information-based loss functions are used in training neural approaches, it seems plausible that refined-information-based losses could similarly be useful whenever higher-order relationships are particularly critical.
> >
> > > Also include some ablations with regards to the more critical components such as a comparison of the benefit (whether for accuracy or computation time) of the block sampling procedure
> >
> > Is this asking for ablation studies on each of the practical contributions we make for the gradient descent algorithm (higher-order Gibbs, AIS, upsampling interactions, etc.)?  The goal of all of these approaches is to get a more accurate estimate of the $\theta^\emptyset$ parameter in order to get a more accurate estimate of the $\eta$ parameters, in a quicker amount of time.  Are you envisioning something like ablating these components individually and recording the time required to get within an epsilon error of the true $\eta$ parameters?  Although we feel that we have intuitively explained why each of these should clearly lead to improved speed, please let us know if this type of experiment is what you have in mind.
> >
> > > a comparison on some toy examples of the differences in selection of interaction collections between the theoretical RI and mutual information (critical).
> >
> > We have added experiments to Appendix B.6 which demonstrate how multiple mutual information (MMI) and refined information (RI) are correlated with each other on a set of 100 random distributions.  We hope this addresses your concerns here, please see our responses to other reviewers about this for a discussion on why there is no reasonable alternative to choosing MMI.
> >
> > Overall, we hope these responses address many of your concerns about the conclusions which are supported by our experiments.  If there are any confusions remaining or any experiments which you feel are necessary to run which we have not yet, please let us know.

---

### Decision · Action_Editor_8pJv · 2026-03-25

**Recommendation:** Accept as is

**Audience:**

Yes

**Audience Explanation:**

Energy-based models form a large class of machine learning models with an active community of practitioners and theoreticians working with them. Their analysis clearly aligns with many readers of TMLR.

**Claims And Evidence:**

Yes

**Claims Explanation:**

As all reviewers agree upon, the mathematics is sound, the extensive experiments (including on real data) support the claims as properly scoped, and the post-revision manuscript appears to have adequately addressed the few clarity concerns. The evidence is accurate and convincing, and the authors delivered appropriate changes after the discussions with reviewers. I therefore recommend publication as is.